# SOLO: SURROGATE ONLINE LEARNING AT ONCE FOR SPIKING NEURAL NETWORKS

## ABSTRACT

Spiking Neural Networks (SNNs) show promise as energy-efficient models inspired by the brain. However, there is a lack of efficient training methods for deep SNNs with online learning rules that mimic biological systems, particularly for deployment on neuromorphic computing substrates. In this paper, we propose Surrogate Online Learning at Once (SOLO) for SNNs, which utilizes several surrogate strategies that could be implemented in a hardware-friendly manner. By exploiting expanded spatial gradient from only the final time step of forward propagation, SOLO achieves low computational complexity while maintaining comparable accruacy and convergence speed. Moreover, the update rule of SOLO takes the simple form of three-factor Hebbian learning, which could enable online on-chip learning. Our experiments on both static and neuromorphic datasets show that SOLO achieves performance comparable to conventional learning algorithms. Furthermore, SOLO is hardware-friendly, offering robustness against device non-idealities and sparse access during write operations to memory devices.

## 1 INTRODUCTION

Spiking neural networks (SNNs) are the next generation of neural network models (Maass, 1997), highlighting their potential for low-power computation and brain-like cognitive function. In SNNs, neurons transmit information through sparse and binary spikes, enabling intuitive hardware design and event-driven computing. From digital neuromorphic computing (NC) chips (Akopyan et al., 2015; Davies et al., 2018; Pei et al., 2019) to mixed-signal NC chips (Pehle et al., 2022; Büchel et al., 2021; Aamir et al., 2018) and neuromorphic devices (Pedretti et al., 2017; Byun et al., 2022), interdisciplinary approaches have contributed to remarkable advancements toward achieving energy-efficient hardware solutions. However, despite the impressive progress in neuromorphic hardware development, the training of SNNs remains a challenging research topic.

Backpropagation, a powerful learning rule in artificial neural networks (ANNs), has been adapted for SNNs. Gradient-based direct training methods with surrogate gradient (SG) (Neftci et al., 2019) are one of the main methods that enable the training of deep SNNs with high performance in large-scale data sets with extremely low latency. (Göltz et al., 2021; Kwon et al., 2020; Lee et al., 2016; Shrestha & Orchard, 2018; Wu et al., 2018; Zheng et al., 2021; Fang et al., 2021a; Han et al., 2020; Deng et al., 2022; Roy et al., 2019) However, the most significant challenge faced by SNNs is the credit assignment problem (Neftci et al., 2019), which arises from representing each spiking neuron as a self-recurrent neural network. During training, they have significant memory requirements that scale with the number of time steps. Furthermore, these methods deviate from the principles of biological online learning, which serve as the learning rule in neuromorphic substrates. (Akopyan et al., 2015; Davies et al., 2018; Pei et al., 2019; Pehle et al., 2022) While they are compatible with event-based inputs and multiple spiking neuron models, they are inefficient in terms of memory and time complexity in practical operation.

In advance, several online training methods (Kag & Saligrama, 2021; Yin et al., 2022; Xiao et al., 2022) and on-chip training methods (Neftci & Indiveri, 2010; Ambrogio et al., 2018; Grübl et al., 2020; Cramer et al., 2022) have been proposed for SNN. However, direct training methods for scalable SNNs often depend on offline off-chip learning due to the credit assignment problem. To improve memory and time complexity, online training methods are typically derived from simplified real-time recurrent learning (RTRL). (Williams & Zipser, 1989; Tallec & Ollivier, 2017; Menick

et al., 2021) While being free from temporal dependence and leveraging local information can be useful strategies for online training, they are not sufficient for large-scale tasks or on-chip implementation. (Zenke & Ganguli, 2018; Trondheim, 2016; Kaiser et al., 2020; Bellec et al., 2020; Bohnstingl et al., 2022)

To achieve scalable, online on-chip training for SNNs, it is important to embed neuromorphic substrates (Zenke & Neftci, 2021) in emerging devices (Kim et al., 2015; Ishii et al., 2019a; Geoffrey W. Burr & Leblebici, 2017; Sebastian et al., 2018), enabling high performance with low latency and maintaining online learning properties. Fortunately, recent studies have shown that these emerging devices can be applied to reproduce complex biological functions, most of which exhibit leaky properties, with low power consumption. (Marković et al., 2020; Kumar et al., 2022; Wang et al., 2017; Xia & Yang, 2019) However, non-idealities in synaptic devices can lead to durability issues and a significant degradation in accuracy when deploying SNN training on neuromorphic substrates. (Gonugondla et al., 2018; Tsai et al., 2018) To address these issues, online on-chip learning algorithms are expected to be co-designed with implementable neuromorphic substrates, taking into account their impact on NC chips. (Gallo et al., 2023; Chen et al., 2018)

In this work, we introduce a novel approach to training spiking neural networks that achieves comparable performance with low latency while maintaining the online learning properties necessary for learning on NCs. We first propose neuron models that include trainable time constant parameters, referred to as the pseudo-Parametric Leaky Integrate-and-Fire (pPLIF) spiking neuron models and pseudo-Parametric leaky integration (pPLI) neuron models. Next, we propose Surrogate Online Learning at Once (SOLO) for SNNs, which utilizes surrogate strategies to perform learning with low computational complexity by exploiting gradient from the final time step. We conduct extensive experiments, demonstrating that SOLO achieves comparable performance on large-scale static and neuromorphic datasets. SOLO could also achieve a convergence speed comparable to backpropagation through time (BPTT) and spatial-temporal backpropagation (STBP) on the CIFAR-10 dataset. Furthermore, SOLO is designed to be hardware-friendly, performing efficient online on-chip learning. We verify the effectiveness of SOLO in online learning contexts. We demonstrate that SOLO can address device durability problems with sparse write access and non-ideality issues of analog computing substrates, including thermal noise and device mismatch.

## 2 PRELIMINARIES

### 2.1 SPIKING NEURON MODELS

**pPLIF neuron models.** To demonstrate SOLO, we base on Leaky Integrate-and-Fire (LIF) neuron models (Gerstner & Kistler, 2002), which are transformed into an iterative expression using the Euler method. The Parametric LIF (PLIF) neuron models (Fang et al., 2021b) were proposed to enhance the performance of SNNs by introducing trainable membrane time constants, which enable them to exhibit LSTM-inspired dynamics. We propose the pseudo-Parametric Leaky Integrate-and-Fire (pPLIF) spiking neuron models to learn the membrane time constants of SNNs. Compared to the Parametric Leaky Integrate and Fire (PLIF) neuron models, the pPLIF neuron models are more hardware-implantable and simplifies the computation of gradient flowing to a trainable membrane time constant $\tau_{mem}^l$ (see details in Supplementary Section A and Supplementary Section B). The membrane time constant $\tau_{mem}^l$ is shared among neurons in the same layer in SNN. We consider simple current models: $I_i^l[t] = \sum W_{ij}^l S_j^{l-1}[t]$ where the subscript $i$ represents the $i$-th neuron and $W_{ij}$ are the weights of neurons $j$ to neurons $i$, without bias. The discrete computational form is:

$$\begin{cases} U_i^l[t] = \beta^l U_i^l[t-1](1 - S_i^l[t-1]) + I_i^l[t] \\ S_i^l[t] = \Theta(U_i^l[t] - \vartheta_{\text{th}}) \end{cases} \tag{1}$$

where $U_i^l$ are the subthreshold membrane potentials of neurons $i$, $\beta^l = 1/(1 + exp(-\tau_{mem}^l))$ is the membrane potential decaying constant at layer $l$, $S_i^l$ are the occurrences of output spikes of neurons $i$ at layer $l$, $\Theta(\cdot)$ is heaviside function, and $\vartheta_{\text{th}}$ is the threshold. The hard reset operation is implemented by multiplying the occurrences of an output spikes of neurons $i$ at time step $t-1$.

## 2.2 Accumulative Neurons

**pPLI mode of accumulative neurons.** Before computing loss with the spike counting method, the output spike count is updated in the spike accumulator. We propose accumulative neurons that can operate in both the integration (I) mode (see details in Supplementary Section A), which is same as spike accumulator, and the pseudo-Parametric leaky-integration (pPLI) mode with a trainable membrane time constant. The operation of the pPLI neuron models are similar to the pPLIF neuron models without firing. It functions as follows:

$$A_i[t] = \beta^A A_i[t-1] + S_i^L[t] \tag{2}$$

where $A_i$ are the membrane potentials of the accumulative neurons $i$, $\beta^A = 1/(1 + exp(-\tau_{mem}^A))$ is the membrane potential decaying constant, and $S_i^L$ is the occurrences of output spikes of neurons $i$ at the last layer $L$.

## 2.3 Conventional Learning Algorithms

Backpropagation Through Time (BPTT) (Neftci et al., 2019) and Spatio-Temporal Backpropagation (STBP) (Wu et al., 2018) are the popular learning algorithms for training SNNs.

**Backpropagation Through Time.** BPTT is a training algorithm originally designed for recurrent neural networks (RNNs), and it has been adapted to interpret SNNs as RNNs for training purposes. As shown in Supplementary Figure 1(a), the loss is computed at each time step, addressing both spatial and temporal errors.

**Spatio-Temporal Backpropagation.** STBP is similar to BPTT, but is designed to leverage spatio-temporal information in SNNs more effectively. As shown in Supplementary Figure 1(b), while the loss is computed at the final time step, the gradient of each time step are propagated through the spike counter, addressing both spatial and temporal errors.

However, both of these methods involve considerable computational intensity due to the necessity of handling intermediate states and gradient for each time step. This computational demand adds complexity to the credit assignment process.

## 3 Surrogate Online Learning at Once

As shown in Figure 1(a), the SOLO follows a forward path through the time steps while considering a backward path only at the final step.

According to Forstmann et al. (2010), the brain exhibits a trade-off between time and accuracy. When inference is performed in a short number of time steps, potential errors can arise. We believe that the information of the accumulative neurons in the final time step could yield the most distinct and clear error value among all given time steps. Moreover, when considering temporal error and retracing through the time steps during backward propagation, the gradient vanishing phenomenon becomes evident. (Ponghiran & Roy, 2022) Given this, we believe that independence from temporal error might not lead to notable challenges. Consequently, the gradient chain of the SOLO relies on a spatial gradient at the final step, reducing the emphasis on backpropagation across both the temporal and spatio-temporal dimensions.

This beneficial behavior is also supported by biological behavior that indicates that neuromodulators (Lovinger et al., 2022; Mei et al., 2022), which can be interpreted as error signals in SNNs, are not constantly secreted but are rather spatially released in specific cycles.

### 3.1 Surrogate Strategies of the SOLO

To realize SOLO, we utilize four surrogate strategies to expand the gradient from the final time step in SOLO. The term "surro" refers to surrogate strategies. (see details in Supplementary Sections B)

**surro1: Wide Range of Boxcar Function as Surrogate Gradient.** Due to the non-differentiable nature of the spiking activation function, we employ the surrogate gradient method. In particular, we utilize the boxcar function, which enables efficient on-chip implementation. We expand the window

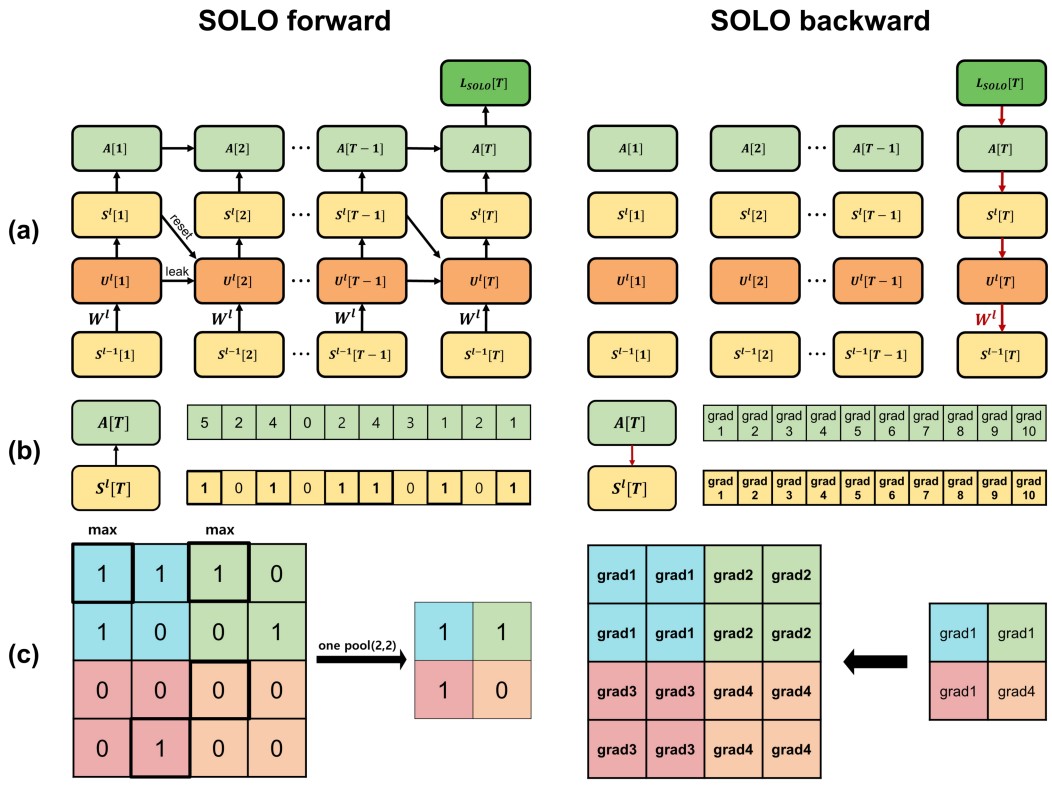

Figure 1: (a) Illustration of the proposed SOLO. (b) Operation of always-on beyond spike accumulation. On the backward path, the gradient is propagated to all elements, ignoring spatial gradient between accumulative neurons and the occurrences of spikes at the last layer. (c) Operation of always-on pooling. On the backward path, the gradient is propagated to all elements. These operations increase the number of candidates that can be updated in the gradient chain.

value to ensure that more neurons fall within the boxcar function.

$$\frac{\partial S[T]}{\partial U[T]} = \Theta'(U[t] - \vartheta_{\text{th}}) \rightarrow \mathbb{1}_U = \Theta(|U[t] - \vartheta_{\text{th}}| < p) \tag{3}$$

where $\mathbb{1}_U$ are the outcomes of the boxcar function applied to the membrane potentials and $p$ is the window value for the range of membrane potentials that allow the flow of gradient.

**surro2: Always-On beyond Spike Accumulation.** During backward propagation, output spikes in the final layer contribute to computing the spatial gradient linked to accumulated neurons. However, as shown in Figure 1(b), we ignore this spatial gradient to achieve an abundant gradient flow, ensuring error propagation across all classes.

$$\frac{\partial A[T]}{\partial S^L[T]} \rightarrow 1 \tag{4}$$

**surro3: Always-on pooling.** During forward propagation, the operation of always-on pooling functions identically to max pooling. However, during backward propagation, it facilitates the flow of gradient through all tensors without considering the maximum value. As shown in Figure 1(c), this operation improves the likelihood of updates.

**surro4: pPTRACE.** From backpropagation through time (BPTT), the gradient of spikes through the time steps can be represented as an eligible potential, which takes the form of leaky integration (LI). (Bellec et al., 2020; Bohnstingl et al., 2022) Expanding on this representation, we propose a pseudo-Parametric Spike Trace (pPTRACE) that replaces the gradient of spikes at the final time step with the computation of eligible spikes through the time steps. This approach effectively increases

the number of spike candidates eligible for weight updates. The computation process for the eligible spikes can be described as follows (see details in Supplementary Sections B):

$$\begin{cases} \tilde{U}^l[T] = k(\alpha^l \tilde{U}^l[T-1] + a^l S^{l-1}[T]) \\ \tilde{S}^l[T] = \Theta(\tilde{U}^l[T] - \tilde{\vartheta}^l_{\text{th}}) \end{cases} \tag{5}$$

where $\tilde{U}^l$ are eligible potentials at layer $l$, $k(\cdot)$ is the clamp function, and $\alpha^l = 1/(1 + exp(-\tau^l_{spk}))$ is the eligible potential decaying constant, $a^l$ is the scaling constant, and $S^{l-1}$ are the occurrences of output spikes of neurons $j$ at layer $l-1$, $\tilde{S}^l[T]$ are eligible spikes at layer $l$, and $\tilde{\vartheta}^l_{\text{th}}$ is the threshold of an eligible spike. $\tau^l_{spk}$, $a^l$ and $\tilde{\vartheta}^l_{\text{th}}$ are trainable parameters and shared in the same layer in SNNs.

To enable the gradient propagation towards trainable parameters, we employ the following techniques:

$$\begin{cases} \frac{\partial U^l_i[T]}{\partial \tilde{S}^l[T]} = \frac{\partial U^l[T]}{\partial \tilde{s}[T]} \frac{\partial \tilde{s}[T]}{\partial \tilde{S}^l[T]} = W^l \\ \frac{\partial \tilde{S}^l[T]}{\partial \tilde{U}^l[T]} = \Theta'(\tilde{U}^l[t] - \tilde{\vartheta}_{\text{th}}) \to \mathbb{1}^l_{\tilde{U}} = \Theta(\tilde{U}[t] - \tilde{\vartheta}_{\text{th}}) \end{cases} \tag{6}$$

where $\tilde{s}(\cdot)$ is the switch function, $\mathbb{1}^l_{\tilde{U}}$ are the outcomes of the heaviside function applied to the eligible potentials.

Finally, we derive eligible spikes that serve as surrogate gradient of spikes for weight updates.

$$\frac{\partial U^l[T]}{\partial W^l} = \mathbb{1}^{l-1}_S = S^{l-1}[T] \to \mathbb{1}^l_{\tilde{S}} = \tilde{S}^l[T] \tag{7}$$

where $\mathbb{1}^{l-1}_S$ are occurrences of spikes at layer $l-1$ and $\mathbb{1}^l_{\tilde{S}}$ are eligible spikes at layer $l$.

## 3.2 DERIVATION OF THE SOLO

We compute the mean squared error (MSE) loss using the firing rate given by $A[T]/T$. The loss function of SOLO is:

$$L_{SOLO} = L_{MSE}(A[T]/T, \hat{y}) \tag{8}$$

where $\hat{y}$ is target label.

Following the chain rule, the gradient of trainable parameter $W^l_{ij}$ and $\tau^l_{mem}$ can be derived as:

$$\begin{cases} \frac{\partial L_{SOLO}}{\partial W^l} = \delta^l[T] \frac{\partial S^l[T]}{\partial U^l[T]} \frac{\partial U^l[T]}{\partial W^l} = \delta^l[T] \mathbb{1}^l_U \mathbb{1}^{l-1}_S \\ \frac{\partial L_{SOLO}}{\partial \tau^l_{mem}} = \delta^l[T] \frac{\partial S^l[T]}{\partial U^l[T]} \frac{\partial U^l[T]}{\partial \tau^l_{mem}} = \delta^l[T] \mathbb{1}^l_U \frac{\partial U^l[T]}{\partial \tau^l_{mem}} \end{cases} \tag{9}$$

where

$$\delta^l[T] = \frac{\partial L_{SOLO}}{\partial S^l[T]} = \begin{cases} A[T]/T - \hat{y} & \text{if } l = L \\ \delta^{l+1}[T] \frac{\partial S^{l+1}[T]}{\partial U^{l+1}[T]} \frac{\partial U^{l+1}[T]}{\partial S^l[T]} = \delta^{l+1}[T] \mathbb{1}^l_U W^l & \text{if } l < L \end{cases} \tag{10}$$

## 3.3 IMPLEMENTATION DETAILS

We consider both versions: SOLO and SOLO$_{\text{et}}$. 'surro 4: pPTRACE' is utilized in SOLO$_{\text{et}}$ (see details in Supplementary Sections A, B).

## 3.4 SIMPLIFIED THREE-FACTOR LEARNING RULES

Surrogate gradient learning can be interpreted as three-factor learning rules. (Neftci et al., 2019) When we specifically outline the gradient of SOLO for the general weight from layer $l$ to layer $l-1$ and connections between any two neurons $i$ and $j$, we have:

$$\frac{\partial L_{SOLO_{\text{et}}}}{\partial W^l_{ij}} = \frac{\partial L_{SOLO_{\text{et}}}}{\partial S^l_i[T]} \frac{\partial S^l_i[T]}{\partial U^l_i[T]} \frac{\partial U^l_i[T]}{\partial W^l_{ij}} \to \delta^l_i[T] \mathbb{1}^l_{U_i} \mathbb{1}^{l-1}_{S_i} \tag{11}$$

where $\delta^l_i$ is the global modulator, $\mathbb{1}^l_{U_i}$ is the surrogate gradient of post-synaptic neurons, which represents post-synaptic activities, and $\mathbb{1}^{l-1}_{S_i}$ is pre-synaptic activities. In SOLO$_{\text{et}}$, $\mathbb{1}^{l-1}_{S_i}$ corresponds

to $\mathbb{1}^l_{\hat{S}_i}$, which represents the tracked presynaptic activities. This is a kind of three-factor learning rule (Frémaux & Gerstner, 2016; Gerstner et al., 2018; Payeur et al., 2021), where local information can be represented as 0 or 1. Binary representation could simplify the computational processes in hardware and offers lower energy consumption.

Neftci et al. (2019); Zenke & Ganguli (2018); Kaiser et al. (2020) utilized a current-based LIF neuron models to represent pre-synaptic activity and eligibility. However, embedding a current-based LIF neuron models on hardware is challenging to implement. (Yang et al., 2020; Davies et al., 2021; Indiveri et al., 2011) In our approach, we utilize a simple LIF neuron models combined with pPTRACE. In pPTRACE, the pre-synaptic activity can be represented through a binarization process. Given its similarity with the dynamics of conventional neuron models, our approach has the potential to ease hardware implementation.

# 4 EXPERIMENTS

We evaluate the effectiveness of the proposed SOLO on the data classification task with the static datasets and neuromorphic datasets. Static datasets we used are MNIST (Lecun et al., 1998), F-MNIST (Xiao et al., 2017), CIFAR-10 (Krizhevsky, 2009), CIFAR-100 (Krizhevsky, 2009), and Tiny Imagenet (Le & Yang, 2015), and neuromorphic datasets are N-MNIST (Orchard et al., 2015), CIFAR10DVS (Amir et al., 2017), and DVSgesture (Li et al., 2017). We use the network architecture (128Ck3-128SRBk5-P2-256SRBk5-skip-P2-512SCBk3-512SRBk5-P2-FC-Voting/10) proposed by AutoSNN (Na et al., 2022). We conduct the experiments with a time step $T$ of 5 for static datasets and 20 for neuromorphic datasets. (See details of training configuration and additional experiments in Supplementary Section C)

Table 1: Performance on static dataset. We evaluate the test accuracy utilizing both pPLI/I mode of accumulative neurons.

| Dataset | Method | Accuracy(%) |
|---|---|---|
| MNIST | BPTT | 99.15/99.27 |
| | STBP | 99.47/99.39 |
| | SOLO | **99.56/99.57** |
| | SOLO$_{et}$ | **99.36/99.32** |
| F-MNIST | BPTT | 93.73/93.69 |
| | STBP | 93.72/93.57 |
| | SOLO | **94.31/94.16** |
| | SOLO$_{et}$ | **93.61/93.30** |
| CIFAR-10 | AutoSNN | 93.15 |
| | BPTT | 90.44/89.7 |
| | STBP | 90.43/89.61 |
| | BPTT (1.0) | 91.98/**92.11** |
| | STBP (1.0) | 90.21/90.71 |
| | SOLO | **91.33/90.55** |
| | SOLO$_{et}$ | **90.18/89.31** |

Table 2: Performance on various dataset. We evaluate the test accuracy utilizing both pPLI/I mode of accumulative neurons.

| Dataset | Method | Accuracy(%) |
|---|---|---|
| N-MNIST | BPTT | 98.48/98.60 |
| | STBP | 98.78/98.34 |
| | SOLO | **93.80**/90.00 |
| | SOLO$_{et}$ | **94.40**/90.40 |
| CIFAR10DVS | AutoSNN | 72.50 |
| | SOLO | **71.30**/32.30 |
| | SOLO$_{et}$ | 57.40/40.60 |
| DVS128gesture | AutoSNN | 96.53 |
| | SOLO | **90.28**/86.81 |
| | SOLO$_{et}$ | 80.21/81.60 |
| CIFAR-100 | AutoSNN | 69.16 |
| | SOLO | **57.86**/56.02 |
| | SOLO$_{et}$ | 55.25/53.69 |
| Tiny ImageNet | AutoSNN | 46.79 |
| | SOLO | **42.76/44.45** |
| | SOLO$_{et}$ | 31.18/31.24 |

## 4.1 PERFORMANCE OF PATTERN RECOGNITION

We conduct experiments on large-scale static datasets and neuromorphic datasets to evaluate the performance of different methods: BPTT, STBP, SOLO and SOLO$_{et}$. In AutoSNN (Na et al., 2022), the authors employed STBP along with a spike regularization technique.

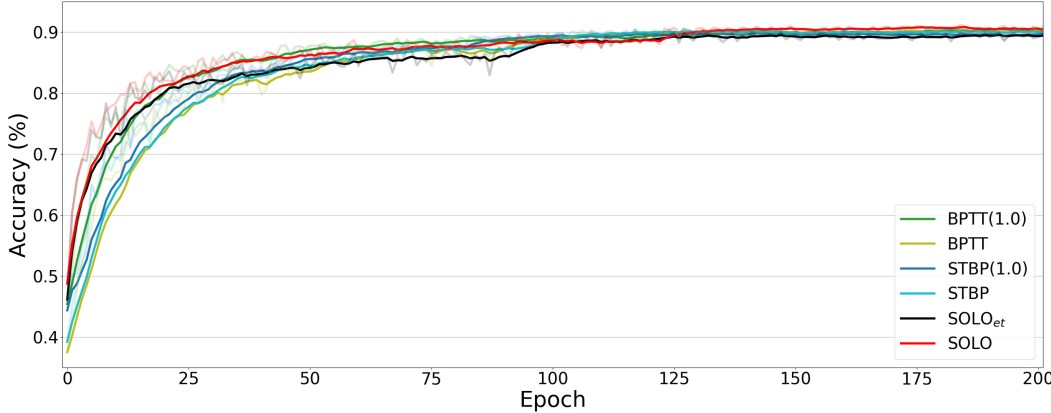

Figure 2: Comparison of the learning curves of SOLO, SOLO_et, BPTT and STBP. The experiments are performed on the CIFAR-10 dataset. The learning curves represent a sequence of validation accuracies. The learning curves of SOLO and SOLO_et closely follow those of BPTT and STBP.

Comparing SOLO and SOLO_et with BPTT, STBP and result of AutoSNN, we note that the best test accuracy drops by less than 2% on static datasets in Table 1 and less than 7% on neuromorphic datasets in Table 2. This indicates that the expanded gradient from the final time step is sufficient to successfully achieve similar accuracy levels. Furthermore, We study the effect of the pPLI mode of accumulative neurons. SNNs with the pPLI mode of accumulative neuron achieve higher test accuracy than I mode on the most cases. This result suggests that the superior performance can be attributed to the pPLI mode of accumulative neurons, which could serve as a regularization effect. (see details in Supplementary Section B)

## 4.2 Comparable Network Convergence Speed

We compare the network convergence speed of SNNs trained with SOLO, SOLO_et, BPTT and STBP on the CIFAR-10 dataset. As shown in Figure 2, the learning curves of SOLO (red line) and SOLO_et (balck line) are closely aligned to BPTT and STBP. The ability of SOLO to closely match these learning curves, even with its simple gradient chain and fast training speed, highlights its efficacy and novelty as a learning algorithm.

Table 3: Comparison of memory and time complexities across various learning rules.

| Method | Memory | Time |
|---|---|---|
| STBP | $O(MLT)$ | $O(NMLT)$ |
| E-prop | $O(NML)$ | $O(NML)$ |
| DECOLLE | $O(1)$ | $O(NM + N_r M)$ |
| SOLO | **O(1)** | **O(NML)** |

Table 4: Performance on CIFAR-10 dataset for different batch sizes over 30 epochs.

| Method | Batch Size | Accuracy (%) |
|---|---|---|
| BPTT | 64/1 | 79.81/61.92 |
| STBP | 64/1 | 80.40/60.66 |
| SOLO | 64/1 | 84.09/**67.64** |
| SOLO_et | 64/1 | 83.45/**65.01** |

## 4.3 Low Memory and Time Complexity

We also examine the memory and time complexity of the proposed SOLO and compare it with other learning rules, as shown in Table 3, where $N$ is the number of input neurons in a layer, $M$ is the number of neurons in a layer, $N_r$ is the number of readout neurons in a layer, $L$ is the number of layers, and $T$ is the size of time steps. In terms of memory complexity, SOLO does not require storing intermediate states for learning, as the states needed for weight updates are readily available from the forward pass. Hence, the memory complexity of SOLO is considered to be $O(1)$. Regarding time complexity, all layers in SOLO need to be sequentially updated from top to bottom. Therefore, the typical time complexity of SOLO is considered to be $O(NML)$. Additionally, as mentioned above,

SOLO utilizes a simple gradient chain with local variables and global errors, allowing a potential reduction in time complexity to $O(L)$ with specific NC substrates.

## 4.4 Performance of Online Training

To evaluate the performance of online training, which involves processing one sample per training step, we performed experiments with a batch size of 1. This approach is consistent with biological learning and learning on NC substrates. We study the performance of different methods on the CIFAR-10 dataset with batch size 1 and compare it with the default batch size of 64 for 30 epochs.

As shown in Table 4, the test accuracy at a batch size of 1 is reduced by less than 25% compared to a batch size of 64. While these results remain less effective than batch learning, they suggest the potential for conducting full online training with the proposed SOLO. Compared to other algorithms, both SOLO and SOLO$_{et}$ demonstrate superior performance in online training.

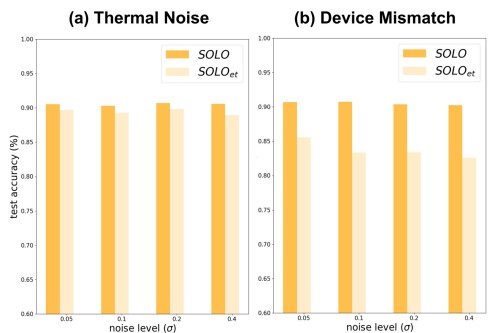

Figure 3: The test accuracy is affected by various hardware-related noise, such as (a) thermal noise and (b) device mismatch. Even as the noise level increases, the test accuracy of SOLO and SOLO$_{et}$ remains relatively stable.

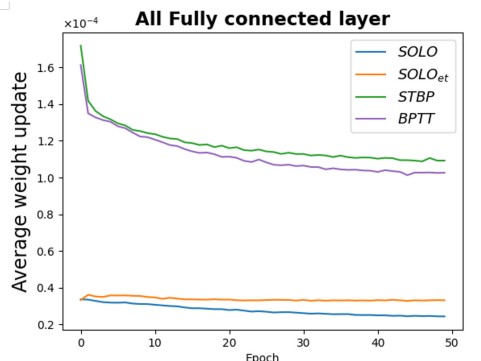

Figure 4: Comparison of the average weight update for each cell in SOLO, SOLO$_{et}$, BPTT, and STBP on the MLP network across all epochs. SOLO and SOLO$_{et}$ exhibit lower values compared to BPTT and STBP.

## 4.5 Robust to Hardware-related Noises

The non-ideality issues associated with NC substrates pose a significant challenge for the deployment of SNNs on mixed-signal NC chips. However, the proposed SOLO offers a potential solution by facilitating on-chip training in a noise-aware manner. (Dutta et al., 2022; Neftci et al., 2016; Ishii et al., 2019b; Lin et al., 2019) In this study, we investigate the effectiveness of SOLO in addressing hardware-related noises with a thermal noise and device update mismatch.

**Thermal Noise.** Thermal noise is an intrinsic source of noise that originates from the neuromorphic circuitry as well as the devices themselves. In our study, we introduce Gaussian noise $N(0, \sigma^2)$ to the input current $I$. The parameter $\sigma$ governs the level of noise, and we adjust it between 0.05 and 0.4 to mimic various noise levels. As shown in Figure 3(a), the test accuracies of SOLO and SOLO$_{et}$ remain stable even as the noise level increases.

**Device Update Mismatch.** During the device update process, the update mismatch causes variations in neuronal and synaptic parameters, leading them to deviate from their target values. To address this issue, we introduce Gaussian noise $N(0, \sigma^2)$ into the gradient computed at each training step. Similar to the experiments conducted for thermal noise, the parameter $\sigma$ governs the level of mismatch, and we adjust it between 0.05 and 0.4 to mimic various noise levels. As shown in Figure 3(b), the test accuracies of SOLO and SOLO$_{et}$ also remain stable even as the noise level increases.

In both noise conditions, SOLO performs better than SOLO$_{et}$. Along with a simple forward-backward computation structure, SOLO exhibits resilience to noise and shows potential for maintaining performance across various noise levels. This suggests that SOLO is expected to effectively handle the noise present in real-world environments.

### 4.6 ROBUST TO HARDWARE-RELATED RELIABILITY

The durability issues associated with NC substrates also pose a significant challenge. As training progresses, weight update signals are repeatedly applied to the synapse devices, which can lead to the accumulation of physical stress and eventual breakdown. (Yu, 2018; Sebastian et al., 2020; Chen, 2016; *et al*, 2021)

However, the proposed SOLO offers a potential solution by employing a low memory access ratio. To verify this, we train a multilayer perceptron (MLP) network on the MNIST dataset for 50 epochs and calculate the average weight update value by summing the values over all batches in one epoch and then dividing by the number of batches. The test accuracies for 50 epochs are as follows: 97.25% for SOLO, 95.96% for $SOLO_{et}$, 97.48% for BPTT, and 97.31% for STBP. As shown in Figure 4, the average weight update of each cell is smaller in both SOLO and $SOLO_{et}$ compared to BPTT and STBP. By minimizing the number of write operations on the synapse memory, SOLO helps alleviate the endurance problems associated with neuromorphic systems, providing a promising approach for long-term reliability in training spiking neural networks.

## 5 CONCLUSION

In this work, we introduce a novel training method called Surrogate Online Learning (SOLO) for spiking neural networks. SOLO employs surrogate strategies, utilizing the expanded spatial gradient from the final time step without relying on a temporal or spatial-temporal gradient through time. This approach resolves the credit assignment problem and substantially reduces computational complexity.

We first conduct a case study on large-scale static experiments that demonstrate the effectiveness of SOLO, yielding levels of accuracy comparable to those of the BPTT and STBP. Additionally, we compare the convergence speed of SOLO to that of conventional methods. We find that SOLO achieves a similar learning curve with efficient operations. Moreover, under online on-chip conditions, we demonstrate that SOLO exhibits robustness against online conditions and various non-ideality issues associated with NC substrates.

In conclusion, the proposed surrogate online learning (SOLO) holds great promise for the efficient training and deployment of spiking neural networks on mixed-signal NC platforms.

## 6 FUTURE WORKS

SOLO demonstrates that competitive learning outcomes can be achieved by focusing only on the information from the final time step. This novel approach and the insights shared in this paper could serve as a milestone for future research on lightweight learning algorithms for SNNs and neuromorphic hardware.

In the near future, we intend to implement SOLO on NC platforms. Furthermore, we also aim to adapt SOLO for spatial-temporal data more sufficiently by utilizing an approach that sparsely performs backward operations. Especially the refinement of $SOLO_{et}$, which incorporates pPTRACE, may be even more suitable for spatial-temporal tasks. We are also investigating its potential for object detection tasks like Su et al. (2023); Kim et al. (2020).

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

## A PRELIMINARY AND SURROGATE STRATEGIES

### A.1 NEURON MODEL

**Parametric leaky integrate-and-fire (PLIF) Neuron Model**

$$\begin{cases} I_i^l[t] = \sum W_{ij}^l S_j^{l-1}[t] \\ U_i^l[t] = (1 - \beta^l)U_i^l[t-1](1 - S_i^l[t-1]) + \beta^l I_i^l[t] \\ S_i^l[t] = \Theta(U_i^l[t] - \vartheta_{\text{th}}) \end{cases} \tag{1}$$

where $I_i^l[t]$ are simple input current models, $W_{ij}$ are the weights from neurons $j$ to neurons $i$ and $S_j^{l-1}$ are the occurrences of output spikes of neurons $j$ at layer $l-1$, $U_i^l$ are the subthreshold membrane potentials of neurons $i$ in $l$ layer, $\beta^l = 1/(1 + exp(-\tau_{mem}^l))$ is the membrane potential decaying constant at layer $l$, $S_i^l$ are the occurrences of output spikes of neurons $i$ at layer $l$, $\Theta(\cdot)$ is heaviside function, and $\vartheta_{th}$ is the threshold. The hard reset operation is implemented by multiplying the occurrences of the output spikes of neurons $i$ at time step $t-1$.

The PLIF neuron model introduces trainable membrane time constants. In its mathematical representation, it employs a weighted sum of the membrane potential and input current. This design enables the model to exhibit LSTM-inspired dynamics. In contrast, the pPLIF neuron model does not introduce the membrane time constants with the input current. As a result, the pPLIF neuron model is more straightforward for hardware implementation.

### A.2 ACCUMULATIVE NEURON

**Integrate(I) mode of Accumulative Neuron**

$$A_i[t] = A_i[t-1] + S_i^L[t] \tag{2}$$

where $A_i$ are the membrane potentials of the accumulative neurons $i$ and $S_i^L$ are the occurrences of the output spikes of neurons $i$ at the last layer $L$.

### A.3 BACKPROPAGATION THROUGH TIME AND THE CONCEPT OF SPIKE TRACE

In the gradient chain of Backpropagation Through Time (BPTT), the spike traces of leaky integrate-and-fire (LIF) neuron models can be represented as follows:

$$\frac{\partial L_{BPTT}}{\partial W^l} = \sum_t \sum_{t' \leq t} \frac{\partial L_{BPTT}}{\partial S[t]} \frac{\partial S^l[t]}{\partial U[t]} \frac{\partial U^l[t]}{\partial U[t-1]} \cdots \frac{\partial U^l[t'-1]}{\partial U[t']} \frac{\partial U^l[t']}{\partial W^l} \tag{3}$$

$$= \sum_t \delta^l[t] \frac{\partial S^l[t]}{\partial U^l[t]} \sum_{t' \leq t} \frac{\partial U^l[t]}{\partial U[t-1]} \cdots \frac{\partial U^l[t'-1]}{\partial U[t']} \frac{\partial U^l[t']}{\partial W^l} \tag{4}$$

$$\simeq \sum_t \delta^l[t] \frac{\partial S^l[t]}{\partial U^l[t]} \sum_{t' \leq t} \beta^{t-t'-1} \frac{\partial U^l[t']}{\partial W^l} \tag{5}$$

$$= \sum_t \delta^l[t] \frac{\partial S^l[t]}{\partial U^l[t]} \sum_{t' \leq t} \beta^{t-t'-1} S^{l-1}[t'] \tag{6}$$

$$= \sum_t \delta^l[t] \frac{\partial S^l[t]}{\partial U^l[t]} \bar{S}^{l-1}[t] \tag{7}$$

where $\beta$ is the membrane potential decaying constant and $\bar{S}^l[t]$ are the spike traces at layer $l$. and $\bar{S}^l[t]$ can be represented as follows:

$$\bar{S}^l[t] = \beta \bar{S}^l[t-1] + S^l[t] \tag{8}$$

$\bar{S}^l[t]$ operate similarly to a specific potential that can be described in a low-pass filter-like manner Bellec et al. (2020). This constitutes the fundamental principle of the eligible trace.

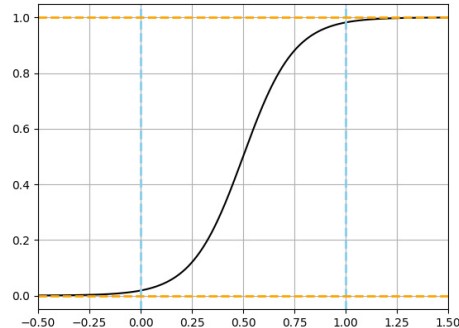

Supplementary Figure 1: Clamp function for eligible potential of pPTRACE.

## A.4 SURRO4 : PPTRACE

We intend to utilize the eligible spike, derived from the concept of the spike trace in Supplementary Equation (8). Given that spike traces can have values exceeding 0, we need a clamp function paired with a threshold to transform the spike trace into a binary representation.

**clamp function for eligible potential**

As shown in Supplementary Figure 1, the clamp function for eligible potentials, denoted as $k(\cdot)$, ensures that the values of eligible potentials range between 0 and 1, centered at (0.5, 0.5). This guarantees that the eligible potentials stay within the desired range. Furthermore, a threshold is utilized to convert the eligible potential into a binary representation, which is referred to as the eligible spike on pPTRACE.

$$k(x) = \frac{1}{1 + e^{-(x-0.5)\cdot 8}} \tag{9}$$

**clamp function for trainable parameters**

The clamp function for trainable parameters, which is a sigmoid function denoted as $\sigma(\cdot)$, ensures that the values of trainable parameters range between 0 and 1. Trainable parameters, denoted as $\theta$, are the eligible potential decaying constant, the scaling constant, and the threshold of an eligible spike.

**swtich function for eligible spike**

The switch function, denoted as $\tilde{s}(\cdot)$, enables the use of eligible spikes as substitutes for the occurrences of spikes on the backward path. While this function produces only a single output variable on the forward path, it ensures that both variables retain a continuous gradient chain on the backward path. This setup ensures the gradient is propagated to both variables, each with a gradient value of 1.

$$\tilde{s}(F, \tilde{F}) = F \tag{10}$$

$$\frac{\partial \tilde{s}(F, \tilde{F})}{\partial F} = \frac{\partial \tilde{s}(F, \tilde{F})}{\partial \tilde{F}} \rightarrow 1 \tag{11}$$

Therefore, we utilize the switch function with eligible spikes as follows:

$$\tilde{s}(S^l[T], \tilde{S}^l[T]) = S^l[T] \tag{12}$$

$$\frac{\partial \tilde{s}(S^l[T], \tilde{S}^l[T])}{\partial S^l[T]} = \frac{\partial \tilde{s}(S^l[T], \tilde{S}^l[T])}{\partial \tilde{S}^l[T]} \rightarrow 1 \tag{13}$$

## A.5 RELATED GRADIENT CHAIN

We introduce an additional gradient chain for trainable membrane constants within SOLO. Relative to the complete gradient chain, SOLO considerably simplifies the gradient computation terms, suggesting its suitability for hardware implementation.

**Towards membrane constant on PLIF neurons**

$$\frac{\partial U_i^l}{\partial \tau_{mem}} = \sigma'(\tau_{mem})U_i^l[t-1](1 - S_i^l[t-1])$$

$$+ \sigma(\tau_{mem})\frac{\partial U_i^l[t-1]}{\partial \tau_{mem}}(1 - S_i^l[t-1]) - \sigma(\tau_{mem})U_i^l[t-1]\frac{\partial(S_i^l[t-1])}{\partial(\sigma(\tau_{mem}))} + \sigma'(\tau_{mem})I_i^l[t] \quad (14)$$

where $\sigma(\cdot)$ is the sigmoid function, utilized as a clamp function to ensure that the value of the membrane constant ranges between 0 and 1. In SOLO, both spatial-temporal and temporal gradients from the previous time step are detached, and the simplified gradient chain can be described as follows:

$$\left[\frac{\partial U_i^l}{\partial \tau_{mem}}\right]_{SOLO} = -\sigma'(\tau_{mem})U_i^l[t-1](1 - S_i^l[t-1]) + \sigma'(\tau_{mem})I_i^l[t] \quad (15)$$

**Towards membrane constant on pPLIF neurons**

$$\frac{\partial U_i^l}{\partial \tau_{mem}} = \sigma'(\tau_{mem})U_i^l[t-1](1 - S_i^l[t-1])$$

$$+ \sigma(\tau_{mem})\frac{\partial U_i^l[t-1]}{\partial \tau_{mem}}(1 - S_i^l[t-1]) - \sigma(\tau_{mem})U_i^l[t-1]\frac{\partial(S_i^l[t-1])}{\partial(\sigma(\tau_{mem}))} \quad (16)$$

As same as above, the gradient chain toward $\tau_{mem}$ can be described as follows:

$$\left[\frac{\partial U_i^l}{\partial \tau_{mem}}\right]_{SOLO} = \sigma'(\tau_{mem})U_i^l[t-1](1 - S_i^l[t-1]) \quad (17)$$

In comparing Supplementary equation (15) to Supplementary equation (17), the gradient chain of pPLIF is simpler than that of PLIF, facilitating its implementation in hardware.

**Towards membrane constant on pPLI mode of accumulative neurons**

$$\frac{\partial A_i[t]}{\partial \tau_{mem}^A} = \sigma'(\tau_{mem}^A)A_i[t-1] + \sigma(\tau_{mem}^A)\frac{\partial A_i[t-1]}{\partial \tau_{mem}^A} \quad (18)$$

As same as above, the gradient chain toward $\tau_{mem}^A$ can be described as follows:

$$\left[\frac{\partial A_i[t]}{\partial \tau_{mem}^A}\right]_{SOLO} = \sigma'(\tau_{mem}^A)A_i[t-1] \quad (19)$$

# B  Surrogate Online Learning at Once

## B.1  Illustration of SOLO$_{ET}$

We present illustration of SOLO in our main paper, Figure 2(a). The illustration of SOLO$_{et}$ is shown in Supplementary Figure 2(c).

## B.2  Derivation of the SOLO$_{ET}$

Following the chain rule, the gradients of trainable parameters $W_{ij}^l$, and $\theta^l$ can be derived as:

$$\begin{cases} \frac{\partial L_{SOLO_{et}}}{\partial W^l} = \delta^l[T]\frac{\partial S^l[T]}{\partial U^l[T]}\frac{\partial U^l[T]}{\partial W^l} = \delta^l[T]\mathbb{1}_U^l\mathbb{1}_{\tilde{S}}^l \\ \frac{\partial L_{SOLO_{et}}}{\partial \theta^l} = \delta^l[T]\frac{\partial S^l[T]}{\partial U^l[T]}\frac{\partial U^l[T]}{\partial \tilde{S}^l[T]}\frac{\partial \tilde{S}^l[T]}{\partial \tilde{U}^l[T]}\frac{\partial \tilde{U}^l[T]}{\partial \theta^l} = \delta^l[T]\mathbb{1}_U^l W_{ij}^l\mathbb{1}_{\tilde{U}}^l\frac{\partial \tilde{U}^l[T]}{\partial \theta^l} \end{cases} \quad (20)$$

**Towards trainable parameters on pPTRACE**

We introduce gradient chain for pPTRACE parameters without spatial-temporal and temporal gradients.

$$\left[\frac{\partial \tilde{U}_i[t]}{\partial \tau_{spk}}\right]_{SOLO} = k'(\sigma(\tau_{spk}^l)\tilde{U}^l[T-1] + aS^{l-1}[T]) \cdot \sigma'(\tau_{spk}^l)\tilde{U}^l[T-1] \tag{21}$$

$$\left[\frac{\partial \tilde{U}_i[t]}{\partial a}\right]_{SOLO} = k'(\sigma(\tau_{spk}^l)\tilde{U}^l[T-1] + aS^{l-1}[T]) \cdot S^{l-1}[T] \tag{22}$$

$$\left[\frac{\partial \tilde{U}_i[t]}{\partial \tilde{\vartheta}_{\text{th}}}\right]_{SOLO} = k'(\sigma(\tau_{spk}^l)\tilde{U}^l[T-1] + aS^{l-1}[T]) \cdot (-\Theta'(\tilde{U}^l[t] - \tilde{\vartheta}_{\text{th}})) \tag{23}$$

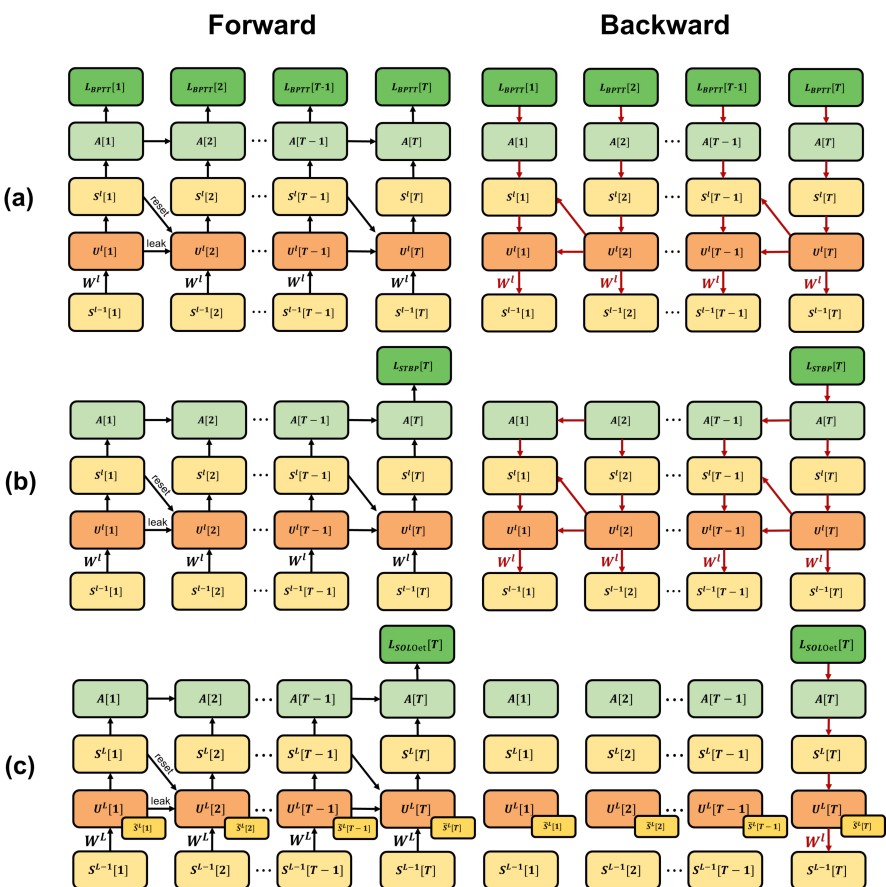

Supplementary Figure 2: Illustration of (a) BPTT, (b) STBP and (c) SOLO$_{\text{et}}$.

### B.3 Comparison of operational characteristics for surro2 and surro3.

As shown in Supplementary Figure 3 (a) and (c), the non-always-on beyond spike accumulation and max pooling determines the spatial gradient based on the occurrence of spikes in a given time step. As the gradient propagates through all time steps, every occurrence of spikes gets the opportunity to contribute to the spatial gradient. More specifically, by referring to Supplementary Figure 3 (e) and (f), we can represent the operational characteristics across different time steps. However, SOLO, which only considers the spatial gradient of the final time step, employs always-on beyond spike accumulation (as shown in Supplementary Figure 3 (b)) and always-on pooling (as shown in Supplementary Figure (d)) to capture the extended potential of the gradient.

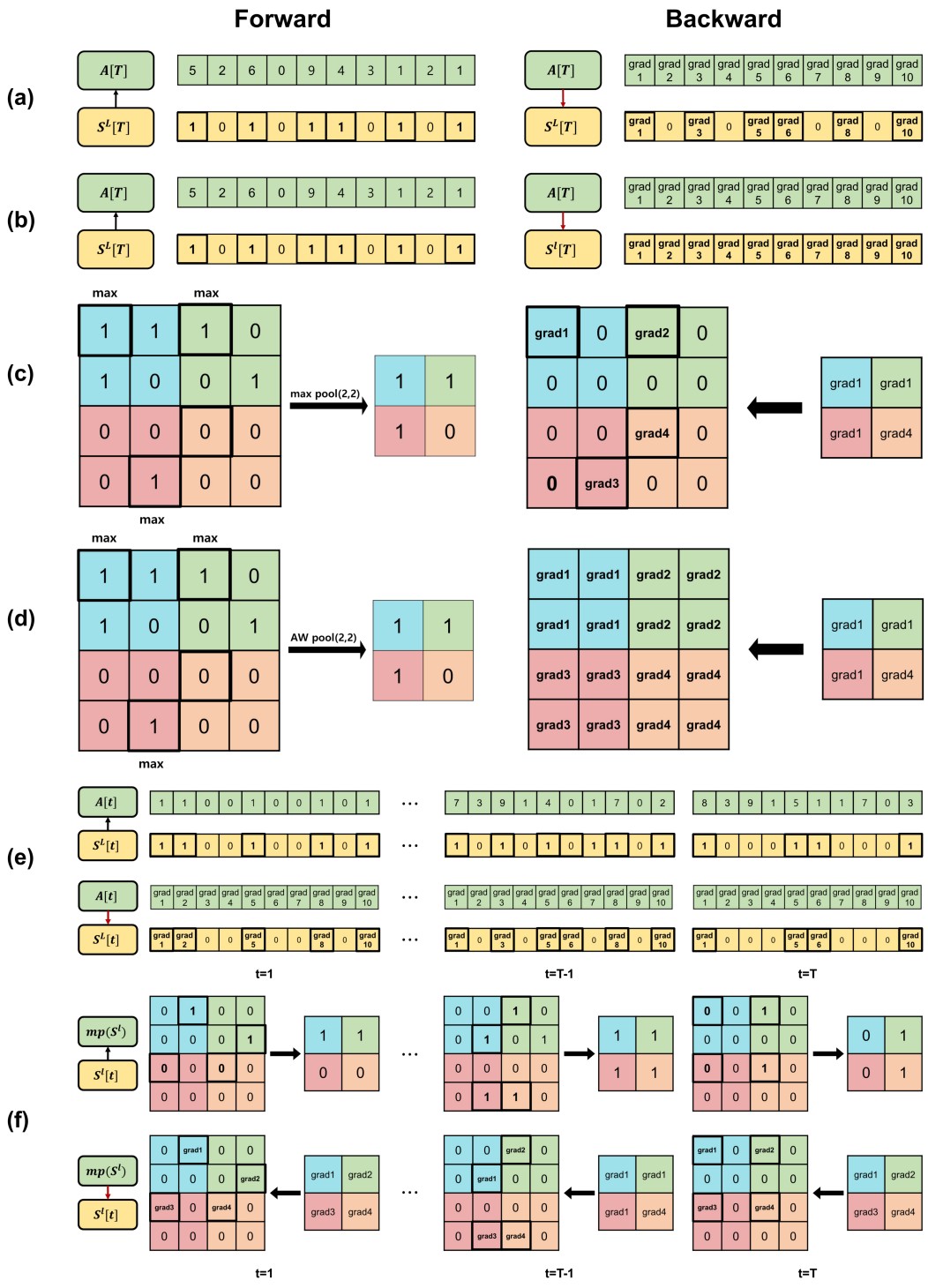

Supplementary Figure 3: (a) non always-on beyond spike accumulation. On the backward path, the gradients are propagated to the occurrences of spikes at the last layer. (b) always-on beyond spike accumulation. On the backward path, the gradients are propagated to all elements. (c) max pooling. On the backward path, the gradients are propagated to the elements with a value of max. (d) always-on pooling. On the backward path, the gradient are propagated to all elements. (e)(f) operational characteristics of non always-on beyond spike accumulation and max pooling across time step. Through this approach, the spatial gradient adjusts across time steps.

## B.4 Ablation Study for surro1: sweeping the window value $p$

We perform image classification on the CIFAR10 dataset, sweeping the window value $p$.

Supplementary Table 1: Results of test accuracy for sweeping $p$ value within SOLO

| $p$ | 0.1 | 0.2 | 0.3 | 0.4 | 0.5 | 0.6 | 0.7 | 0.8 | 0.9 | 1.0 |
|---|---|---|---|---|---|---|---|---|---|---|
| accuracy | 15.85 | 33.87 | 62.59 | 72.73 | **87.25** | 89.73 | 90.12 | 89.99 | 90.96 | **91.33** |

## B.5 Ablation Study for Surrogate Strategies

We conduct an ablation study to evaluate the effectiveness of the proposed surrogate strategies. By analyzing their impact on accuracy, we investigate various configurations within SOLO, specifically focusing on surro1 (wide range of boxcar function, $p$), surro2 (always-on beyond spike accumulation, AW), and surro3 (always-on pooling, AW), as well as different modes of accumulative neuron activation. The specific details of surro2 and surro3 are illustrated in Supplementary Figure 3. We perform image classification on the CIFAR-10 dataset under various configurations. As shown in Supplementary Table 2, training with a combination of always-on beyond spike accumulation, always-on pooling, and the PLI mode of accumulative neuron achieves the highest test accuracy.

Supplementary Table 2: Ablation Study for Surrogate Strategies. We evaluate the test accuracy of SOLO utilizing both pPLI mode/I mode of accumulative neurons.

| Dataset | $p$ | Pooling | nonAW | AW |
|---|---|---|---|---|
| CIFAR-10 | 0.5 | Maxpooling | 87.82/85.97 | 86.80/87.15 |
| | 1.0 | Maxpooling | 90.8/88.82 | 89.45/90.69 |
| | 0.5 | Avgpooling | 90.01/89.60 | 90.47/89.99 |
| | 1.0 | Avgpooling | 90.99/90.96 | 90.88/**91.02** |
| | 0.5 | AWpooling | 87.25/80.50 | 87.25/88.13 |
| | 1.0 | AWpooling | 90.38/90.24 | **91.33/90.55** |

## B.6 Ablation Study for Neuron Models

We conduct an ablation study to evaluate the effectiveness of the proposed pPLIF neuron models. By analyzing their impact on accuracy, we investigate pPLI neuron models and PLIF neuron models within different algorithms. We perform image classification on the CIFAR10 dataset under different neuron models.

As shown in Supplementary Table 3, training with the pPLIF neuron model, even with its simpler gradient chain, yields better results than the PLIF neuron. BPTT (1.0) and STBP (1.0) indicate that the value of $p$ is 1.0.

Supplementary Table 3: Ablation Study for pPLIF/PLIF neuron models. We evaluate the test accuracy of BPTT, STBP, SOLO and SOLO$_{et}$ utilizing pPLI mode of accumulative neuorns.

| Dataset | Algorithm | PLIF | pPLIF |
|---|---|---|---|
| CIFAR-10 | BPTT (1.0) | 90.46 | 90.44 |
| | STBP (1.0) | 90.35 | 90.43 |
| | SOLO | 90.46 | **91.33** |
| | SOLO$_{et}$ | 89.63 | **90.18** |

When combined with the MSE loss, the PLI mode of accumulative neurons could consistently exhibit a regularization effect. When leaky potentials are translated into firing rates (normalized by time step), they never reach 1. Thus, computing the one-hot coded label with the MSE loss always leads to an error. We derive this effect from Temporal Efficient Training Deng et al. (2022).

## B.8    RELATIVE FLOPs OF SOLO

SOLO detaches the gradient chain of spikes and membrane potential through time, maximizing computational efficiency for backpropagation. Generally, backpropagation requires twice the number of operations as forward propagation during matrix multiplication. When applying the SOLO method to an SNN with a timestep length of T, the operations required for backpropagation decrease by a factor of 1/T compared to BPTT. The backpropagation path needs twice the forward path because it needs computation for updating weight and propagating backward. As a result, it is anticipated that the total operations will reduce to $\frac{(FLOPs\,for\,SOLO)}{(FLOPs\,for\,BPTT)} = \frac{1+\frac{2}{T}}{3}$ times. Although the actual runtime of a program is closely related to memory access and does not directly correlate with FLOPs, it has been observed that employing the SOLO method accelerates training (refer to Supplementary 3.7).

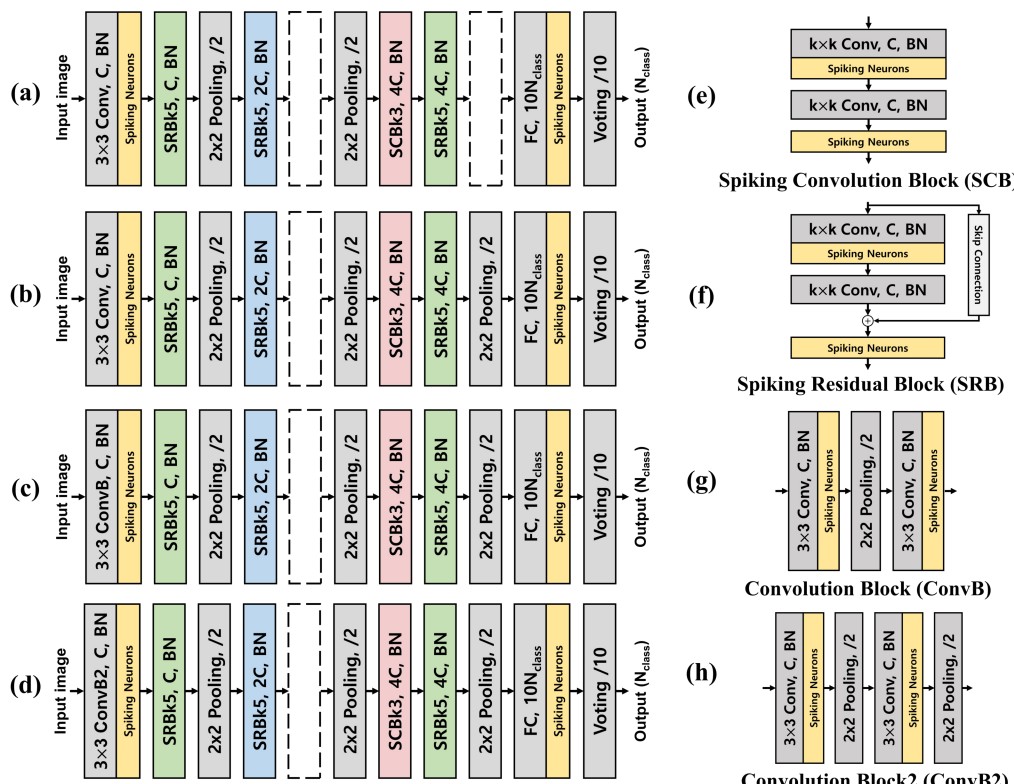

Supplementary Figure 4: (a) architecture for MNIST, F-MNIST, N-MNIST. (b) architecture for CIFAR-10 and CIFAR-100. (c) architecture for Tiny Imagenet. (d) architecture for CIFAR10DVS and DVSgesture. These architectures are searched from AutoSNN. C is channel length, and BN is batch normalization. (e) spiking convolution block (SCB). (f) spiking residual block (SRB) with skip connection (g) convolution block used in (c). (h) convolution block2 used in (d)

## C EXPERIMENTS

### C.1 DATASETS AND DATA AUGUMENTATION

We conduct experiments on static datasets and neuromorphic datasets. MNIST and F-MNIST include images with a resolution of 32×32 and 1 channel, composed of 55,000 training data, 5,000 validation data, and 10,000 test data. CIFAR-10 and CIFAR-100 include images with a resolution of 32×32 and 3 channels (RGB channels), composed of 45,000 training data, 5,000 validation data, and 10,000 test data. Tiny Imagenet includes images with a resolution of 64×64 and 3 channels (RGB channels), composed of 100,000 training data, 10,000 validation data, and 10,000 test data. N-MNIST includes images with a resolution of 34×34 and 2 channels (polarity channels), composed of 55,000 training data, almost 5,000 validation data, and 10,000 test data. CIFAR10DVS includes images with a resolution of 128x128 and 2 channels (polarity channels), composed of 8,900 training data, 100 validation data, and 1,000 test data. DVSgesture includes images with a resolution of 128x128 and 2 channels (polarity channels), composed of 1,176 training data and 288 test data.

We normalize the inputs using the global mean and standard deviation and employ data augmentation techniques like random cropping, horizontal flipping, and cutout DeVries & Taylor (2017). For each time step, the first layer of the SNNs receives pixel values directly, serving as a real-valued input current.

### C.2 ARCHITECTURES

We leverage the architecture obtained from AutoSNN, which is a neural architecture search (NAS) technique specifically for spiking neural networks. The architecture is searched in the SNNs search space Na et al. (2022). The architectures are illustrated in Supplementary Figure 4.

### C.3 HYPERPARAMETER SETTING

Supplementary Table 4: Hyperparameters on SOLO

| param | Initial Value | param | Initial Value |
|---|---|---|---|
| C | 128 | $\tau^A_{acc,init}$ | 2.0 |
| $\tau^l_{mem,init}$ | 2.0 | $\tau^l_{spk,init}$ | 1.0 |
| $\vartheta_{th}$ | 0.5 | $\tilde{\vartheta}_{th}$ | -0.35 |
| $p$ | 1.0 | $a$ | -0.8 |

The hyperparameters are as shown in Supplementary Table 4. The time steps, $T$, for forward propagation have a value of 5 for static datasets and 20 for neuromorphic datasets.

### C.4 TRAINING CONFIGURATION FOR PATTERN RECOGNITION

**MNIST, F-MNIST, and CIFAR-10 datasets.** We train SNNs for 300 epochs using the Adam optimizer. The initial learning rate for Adam is set to 0.0001. The learning rates decay with the ReduceLROnPlateau scheduler with factor 0.1 and patience 10. We train with batch sizes of 64 on an NVIDIA A100 Tensor Core GPU with 40 GB of memory.

**CIFAR-100 and Tiny Imagenet datasets.** We train SNNs for 300 epochs using the Adam optimizer. The initial learning rate for Adam is set to 0.00001. The weight decay is set to $2 \times 10^{-5}$ on Tiny Imagenet, and no dropout is applied. The learning rates decay with the ReduceLROnPlateau scheduler with factor 0.1 and patience 10. We train with batch sizes of 64.

**N-MNIST datasets.** We train SNNs for 300 epochs using the Adam optimizer. The initial learning rate for Adam is set to 0.00005. The learning rates decay with the ReduceLROnPlateau scheduler with factor 0.1 and patience 10. We train with batch sizes of 64.

**CIFAR10DVS datasets.** We train SNNs for 300 epochs using the Adam optimizer. The initial learning rate for Adam is set to 0.0001. The weight decay is set to $1 \times 10^{-5}$, and no dropout

is applied. The learning rates decay with the ReduceLROnPlateau scheduler with factor 0.1 and patience 10. We train with batch sizes of 64.

**DVSgesture datasets.** We train SNNs for 300 epochs using the Adam optimizer. The initial learning rate for Adam is set to 0.0001. The weight decay is set to $1 \times 10^{-5}$, and no dropout is applied. The learning rates decay with the cosine annealing learning rate schedular. We train with batch sizes of 16.

## C.5 MEASUREMENT OF RUN TIME AND GPU MEMORY ALLOCATION ON CIFAR-10

Supplementary Table 5: Measurement of Run Time and Allocated GPU Memory on CIFAR-10 for different learning algorithms

| Method | Run Time | GPU memory |
|---|---|---|
| BPTT | 4 h 10 m | 4.32GB |
| STBP | 4 h 7 m | 3.94GB |
| SOLO | **2 h 13 m** | **1.15GB** |
| SOLO$_{et}$ | 3 h 6 m | 1.99GB |

As shown in Supplementary Table 6, we measure the runtime and GPU memory allocation of each learning algorithm on the CIFAR-10 datasets. To ensure consistent measurement, each GPU is dedicated to executing only one task at a time. As a result, SOLO demonstrates faster completion of the learning process and lower memory usage compared to BPTT and STBP.

## C.6 RESULTS OF SOLO UTILIZAING VARIOUS ARCHITECTURES

Supplementary Table 6: Results of SOLO utilizaing Various architecture

| Dataset | Architecture | STBP | SOLO |
|---|---|---|---|
| CIFAR-10 | CIFARNet-Wu (Wu et al., 2019) | 89.99/90.22 | 90.84/89.00 |
| | CIFARNet-Fang (Fang et al., 2021) | 88.34/81.39 | 89.57/89.30 |
| | ResNet11-Lee (Lee et al., 2020) | 89.26/80.99 | 90.78/90.85 |
| | ResNet19-Zheng (Zheng et al., 2020) | 89.14/89.31 | 91.10/91.24 |
| CIFAR-100 | ResNet19-Zheng (Zheng et al., 2020) | - | **65.72/64.12** |

## C.7 TRAINING CONFIGURATION OF ONLINE TRAINING WITH BATCH SIZE 1

We train SNNs for 30 epochs using the Adam optimizer. The initial learning rate for Adam is set to 0.0001 for batch sizes of 64 and 0.0001/64 for batch sizes of 1. The learning rates decay with the ReduceLROnPlateau scheduler with factor 0.1 and patience 10.

## C.8 RESULT OF ONLINE TRAINING WITH PRE-TRAINED WEIGHT

Table 7: Performance on CIFAR-10 for SOLO/SOLO$_{et}$ with different batch sizes for 30 epochs. Additionally, the results using the pretrained trainable parameters from batch size 64 as initial values are also included.

| Method | Batch | Acc (%) |
|---|---|---|
| BPTT (pretrained) | 1 | 78.45 |
| STBP (pretrained) | 1 | 77.27 |
| SOLO (pretrained) | 1 | **83.76** |
| SOLO$_{et}$ (pretrained) | 1 | **81.88** |

Supplementary Table 8: Performance of Training with Hardware-Related Noise on CIFAR-10. We describe the test accuracy of accumulative neurons in pPLI mode with 5 random seed.

| Related Noise | Method | $\sigma$=0.05 | $\sigma$=0.1 | $\sigma$=0.2 | $\sigma$=0.4 |
|---|---|---|---|---|---|
| Thermal Noise | SOLO | 90.51±0.32 | 90.30±0.63 | 90.68±0.30 | 90.56±0.30 |
| | SOLO$_{et}$ | 89.69±0.50 | 89.30±0.55 | 89.80±0.67 | 88.93±0.48 |
| Device Miss Match | SOLO | 90.69±0.30 | 90.71±0.36 | 90.38±0.43 | 90.24±0.52 |
| | SOLO$_{et}$ | 85.57±0.39 | 83.34±0.32 | 83.36±0.69 | 82.57±0.62 |

As shown in Supplementary Table 7, we describe test accuracy as influenced by the level of noise. We train SNNs for 300 epochs using the Adam optimizer in a noisy configuration with five random seeds. The initial learning rate for Adam is set to 0.0001. The learning rates decay with the ReduceLROnPlateau scheduler with factor 0.1 and patience 10.

## C.10 Training Configuration and Results of Training on Hardware-Related Reliability

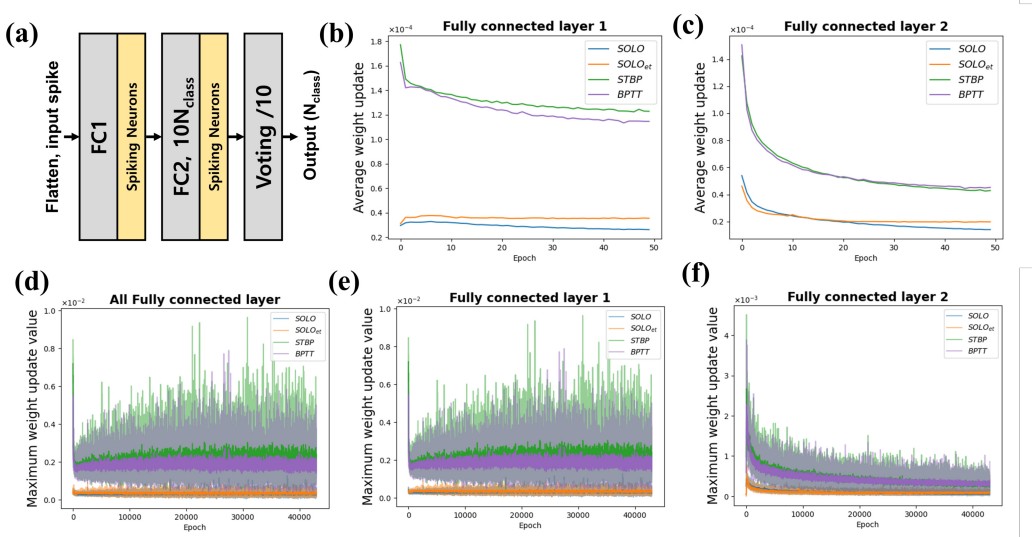

Supplementary Figure 5: MLP architecture for experiment of Training on Hardware-Related Reliability.

As shown in Supplementary Figure 5 (a), we utilize a multi-layer perceptron (MLP) network in this experiment, which consists of the following layers: fully connected layer 1 (FC1), fully connected layer 2 (FC2), and a boost layer. We convert the normalized pixel values into spikes utilizing a spike generator Eshraghian et al. (2023). The FC1 layer connects Input (784 neurons) to Hidden1 (1000 neurons), the FC2 layer connects Hidden1 (1000 neurons) to Hidden2 (100 neurons), and the Boost layer connects Hidden2 (100 neurons) to Output (10 neurons) with average pooling. We train SNNs for 50 epochs using the Adam optimizer. The initial learning rate for Adam is set to 0.0001 for batch sizes of 64.

The graphs (b) and (c), respectively, represent the average weight update values in the FC1 layer and FC2 layer. Both the SOLO and SOLO$_{et}$ algorithms show smaller average update values per cell compared to BPTT and STBP, indicating fewer write operations performed on the synapse memory.

The graphs (d), (e), and (f) represent the maximum values updated in a single cell for each layer, respectively. These measurements were taken for all layers in FC1 and FC2, only in the FC1 layer and only in the FC2 layer. In these graphs, the transparent lines represent the actual values, while the

solid lines represent the values when using the exponential moving average (EMA) with an alpha of 0.1. In all three graphs, it can be observed that in the case of SOLO and SOLO$_{et}$, the maximum update value in a single cell is smaller compared to BPTT or STBP, which makes it more hardware-friendly by reducing the amount of updates required on analog hardware.

## C.11 CROSS BAR ARRAY APPLICATION OF SOLO FOR ANALOG SYNAPSES

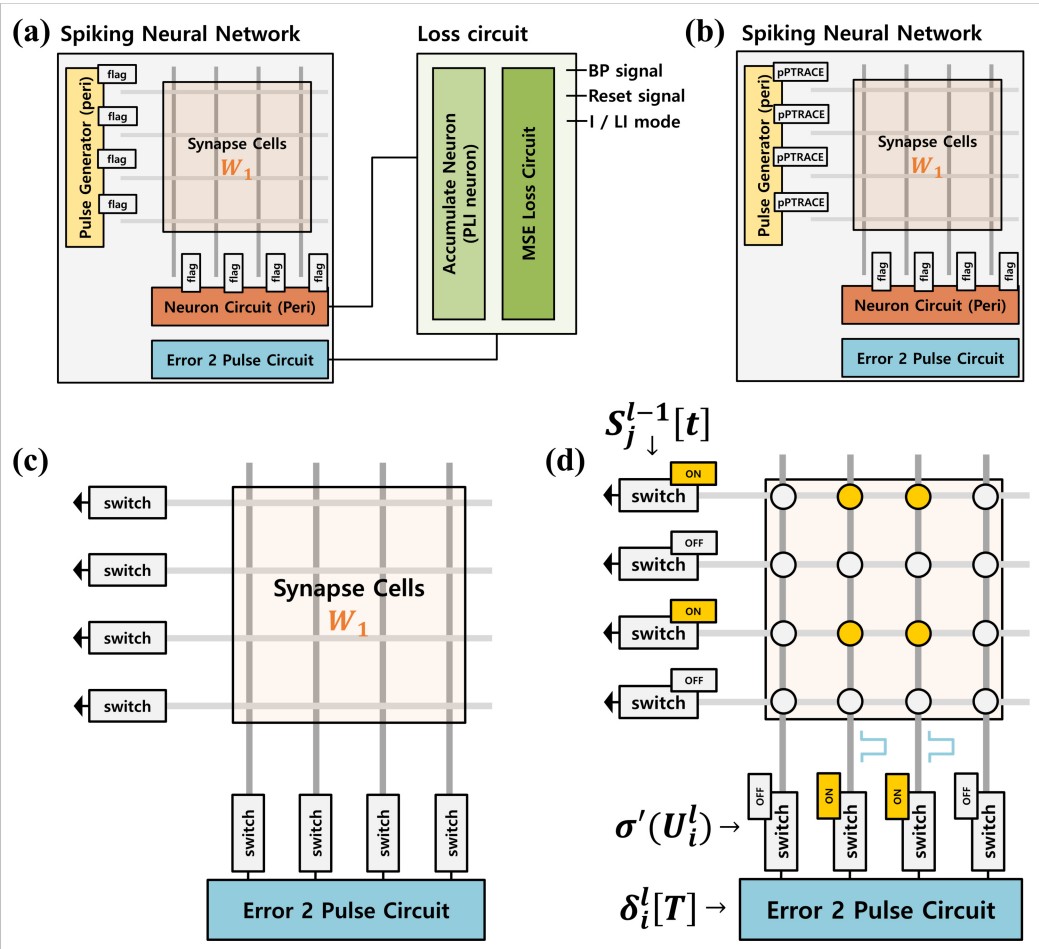

Supplementary Figure 6: Cross Bar Array(CBA) application for analog synapses. (a) Entire spiking neural network layout for SOLO. This example illustrates the full setup of a spiking neural network, incorporating a pulse generator, neuron circuit and error2pulse circuit as peripheral components, centralized around analog CBA for synaptic connections, and a loss circuit. Local memory, labeled as 'flag', captures and represents the states of spikes and the surrogate gradient of membrane potential at the final time step. (b) Spiking neural network layout for SOLO$_{et}$ with pPTRACE. In SOLO$_{et}$ configuration, a pPTRACE circuit is attached to the pulse generator, replacing the spike flag. The pPTRACE circuit also outputs binary values through an Leaky-Integrate operation, similar to operation of neuron circuit. (c) During weight update phase, operational elements carrying binary values could be represented as switches within the peripheral circuit, thereby simplifying the hardware implementation. (d) During weight updates, the rows of CBA could be selected to indicate the occurrence of spikes at the previous layer, while the columns of CBA could be selected to indicate the surrogate gradient of the membrane potential at the current layer. The delta value could be converted into pulses via the error2pulse circuit before applied to the CBA.

$$\left[\frac{\partial L}{\partial W^l}\right]_{SOLO} = \frac{\partial L}{\partial S^{l+1}[T]} \frac{\partial S^{l+1}[T]}{\partial U^{l+1}[T]} \frac{\partial U^{l+1}[T]}{\partial W^l} \tag{24}$$

$$\left[\frac{\partial L}{\partial W^i}\right]_{DSR} = \frac{\partial L}{\partial o^i} \frac{\partial o^i}{\partial W^i} \text{ where } o^i[N] = \frac{\sum_{n=1}^{N} \lambda_i^{N-n} S^i[n]}{\sum_{n=1}^{N} \lambda_i^{N-n} \Delta t} \tag{25}$$

$$\left[\frac{\partial L}{\partial W^l}\right]_{OTTT} = \sum_{t=1}^{T} \frac{\partial L}{\partial S^{l+1}[t]} \frac{\partial S^{l+1}[t]}{\partial U^{l+1}[t]} \sum_{\tau \le t} \lambda^{t-\tau} \frac{\partial U^{l+1}[\tau]}{\partial W^l} \tag{26}$$

$$\left[\frac{\partial L}{\partial W^l}\right]_{SLTT} = \sum_{t=1}^{T} \frac{\partial L}{\partial U^{l+1}[t]} \frac{\partial U^{l+1}[t]}{\partial S^l[t]} \frac{\partial S^l[t]}{\partial U^l[t]} S^{l-1}[t] \tag{27}$$

$$\left[\frac{\partial L[t]}{\partial W}\right]_{FPTT} = \frac{\partial L[t]}{\partial \hat{y}[t]} \frac{\partial \hat{y}[t]}{\partial S[t]} \frac{\partial S[t]}{\partial U[t]} \frac{\partial U[t]}{\partial W} \text{ where } \hat{y} \text{ is a prediction} \tag{28}$$

Supplementary Table 9: Comparison of Recent Efficient Training Methods for SNNs. TG indicates the presence of Temporal Gradient

| Reference | Algorithms | Architecture | CIFAR-10 | CIFAR10DVS | TG | Time step for BP | Additional computing components |
|---|---|---|---|---|---|---|---|
| proposed | SOLO | AutoSNN (pPLIF) | 91.33% (T=5) | 71.30% (T=20) | x | Final time step | - |
| proposed | SOLO$_{et}$ | AutoSNN (pPLIF) | 90.18% (T=5) | 57.40% (T=20) | x | Final time step | pPTRACE |
| Meng et al. (2023b) | DSR | VGG-11 | 77.27% (T=20) | 95.40% (T=20) | x | Final time step | Spike representation |
| Wu et al. (2019) | STBP | CIFARNet-Wu | 90.53% (T=8) | 60.50% (T=20) | o | Every time step | NeuNorm |
| Na et al. (2022) | STBP | AutoSNN (PLIF) | 93.15% (T=8) | 72.50% (T=20) | o | Every time step | Spike regularization |
| Bellec et al. (2020) | e-prop | RSNN (ALIF) | - | - | x | Every time step | Eligibility trace |
| Xiao et al. (2022) | OTTT$_a$ | VGG (sWS) | 93.58% (T=6) | 76.31% (T=20) | x | Every time step | Presynaptic activity |
| | OTTT$_o$ | VGG (sWS) | 93.10% (T=6) | 77.10% (T=20) | x | Every time step (stepwise update) | Presynaptic activity |
| Meng et al. (2023a) | SLTT | ResNet-18 | 94.59% (T=6) | - | x | Every time step | - |
| | SLTT | VGG-11 | - | 77.30% (T=10) | x | Every time step | - |
| | SLTT-K | VGG-11 | - | 76.70% (T=10) | x | K time step (K=2) | Random sampling for K |
| Yin et al. (2022) | FPTT | LTC-RSNNs | - | 73.20% (T=20) | x | Every time step (stepwise update) | additional weight and loss |
| | FPTT-K | LTC-RSNNs | - | - | x | K-truncated time step (stepwise update) | additional weight and loss |

**STBP** STBP serves as the baseline training algorithm for SOLO. STBP employs backpropagation at every time step with temporal gradients. On the AutoSNN works, STBP utilizes additional computational components such as spike regularization (Pellegrini et al., 2021) to improve the training performance. The spike regularization method should require memory for the spikes at every time step and incorporate an additional computation in the loss term.

**SOLO and SOLO$_{et}$** The proposed SOLO employs backpropagation at final time step without temporal gradients. SOLO$_{et}$ requires additional computation for pPTRACE.

**Differentiation on Spike Representation (DSR)** DSR employs backpropagation at final time step without temporal gradients. However, as a spike representation method, it operates over a long time steps and performs additional computations to represent spikes from all time steps at the final time step. So DSR should require memory for the spikes from all time steps.

**e-prop** Utilizing RSNN and ALIF, e-prop employs backpropagation at every time step without temporal gradients. e-prop should require additional memory and computation for 'Eligibility trace'. When using ALIF neurons, the computation for the eligibility trace are complex, making it challenging to apply to neuromorphic hardware, and it requires memory to store the eligibility trace.

**OTTT$_a$ and OTTT$_o$** Utilizing VGG networks with sWS (scaled weight standardization), both OTTT methods employ backpropagation at every time step without temporal gradients. OTTT$_a$ requires additional memory and computation for 'Presynaptic activity' and accumulating gradients. OTTT$_o$ imposes a reduced memory burden due to its stepwise updating approach.

**SLTT and SLTT-K** SLTT employs backpropagation at every time step, ignoring temporal gradients to reduce memory/time complexity. SLTT-K employs backpropagate at K randomly sampled time steps out of the entire sequence.

**FPTT and FPTT-K** Utilizing RSNN and LTC neuron model (Hasani et al., 2020), FPTT employs backpropagation at every time step without temporal gradients. FPTT requires additional memory

and computation for two types of weights and losses that rely on past information. FPTT-K divides the entire time step into K time windows, requiring memory to accumulate the losses within each window.

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
