# OpenReview forum: "SOLO: Surrogate Online Learning at Once for Spiking Neural Networks"
_ICLR.cc/2024/Conference — Submitted to ICLR 2024_

### Official Review · Reviewer_Ev3G · 2023-10-25

**Soundness:** 2 fair
**Presentation:** 2 fair
**Contribution:** 2 fair
**Rating:** 3
**Confidence:** 3

**Summary:**

The paper presents a novel training method called SOLO, which uses surrogate strategies to perform end-to-end learning with low computational complexity. It is easy to implement on neuromorphic hardware and is evaluated on various static and neuromorphic datasets. The method is compared with existing methods like BPTT, STBP, E-prop, and DECOLLE. The paper also demonstrates SOLO's robustness to hardware-related noises and reliability issues, making it suitable for deployment on neuromorphic computing substrates.

**Strengths:**

On-chip single-time backpropagation: SOLO is a surrogate online learning method that trains deep SNNs end-to-end using spatial gradients and surrogate strategies to reduce computational complexity and memory requirements. It also introduces a simple three-factor learning rule for online on-chip learning on neuromorphic hardware.



Hardware awareness: This algorithm considers too many compatibility issues between neuromorphic computing and SNN. It introduces neuron models like pPLIF and pPLI and uses hardware-friendly surrogate strategies like boxcar function and always-on pooling. The evaluation is done given the hardware-related noise.

**Weaknesses:**

The paper lacks clear theoretical justification for the proposed SOLO method, relying on empirical results and biological plausibility without mathematical analysis or proof of convergence.



Unfair comparison: The paper compares SOLO with offline methods like BPTT and STBP but does not compare it with the newest online methods like OTTT (Xiao et al., NeurIPS 2022), SpikeRepresentation (Meng et al., CVPR 2022), and so on.



Lack of clarity: Some of the mathematical expressions lack proper definition and notation. I am confused by some details.



Minor: the citation is not proper in the content. I think the author should use ‘\citep{}’ instead of ‘\cite{}’ most of the time.

**Questions:**

For equation 6, why is there item $\theta(U^\sim[t]-\theta^\sim_{th})$ rather than $\theta(abs(U^\sim[t]-\theta^\sim_{th})<p)$.



Why does pTRACE need a clamp function $k$? I think equation (5) really ensembles the proposed pPLI (equation (2)). Why don’t you simply use pPLI as a surrogate?



Please point out the difference between the current proposed SOTA online training methods and propagate-only-once training methods. Examples are OTTT (Xiao et al., NeurIPS 2022) and spike representation (Meng et al., CVPR 2022).



How do we implement SOLO on a neuromorphic platform when it has a float-point derivation?

[1] Xiao et al., Online Training Through Time for Spiking Neural Networks, NeurIPS 2022
[2] Meng et al., Training High-Performance Low-Latency Spiking Neural Networks by Differentiation on Spike Representation, CVPR 2022

---

> ### Author Response · Authors · 2023-11-17
> **Initial Response to Reviewer Ev3G**
>
> First and foremost, I would like to express my gratitude for your insightful comments and suggestions.
>
> Please check above 'Initial Response to Reviewer Comments'.
>
> 1. definition and notation
>
>  We have endeavored to express the 'lack of proper definition and notation' issue within the equations as much as possible. Should there be any elements that were not adequately addressed, we have explained the relevant notations in Appendix A.1.~B.2..
>
> 2. citation method
>
> We implement the citation method you suggested. We are grateful for your guidance on this matter.
>
> 3. Equation 6 and the clamp function of pPTRACE
>
> We have explored both scenarios and found that $\Theta(\tilde{U[t]}-\tilde{\theta_{th}})$ is more suitable. pPTRACE is ultimately proposed to output binary values named eligible spikes. As seen in Appendix A.4, pPTRACE is explained along with the clamp function, k, and the relevant information is present. The rationale behind using the clamp function is as follows:
>
>  3-1. As forward propagation occurs, the generated spikes are added to the eligible potential. Unlike neurons, pPTRACE does not reset after a certain event (in the case of neurons, a spike), allowing it to accumulate increasing values. We need a limit range for binarization, hence the necessity for a clamp function (see Appendix Figure 1).
>
>  3-2. While torch.clamp() could have been used to set a limit range, it is unable to create a gradient, thus making it ineffective in generating a gradient that flows with the time constant of pPTRACE.
>
> Because pPTRACE doesn’t have any threshold and just accumulates potential, the Heaviside function is more suitable than a box range. Moreover, the Heaviside function is also suitable for hardware implementation as it can be implemented with just one comparator.
>
> The pPLI neuron acts as a spike counter in SOLO, counting the network's output spikes, which is a completely different role from pPTRACE. Depending on the mode of leaky integration or integration, the pPLI neuron counts (or accumulates) spikes and is utilized during loss calculation.
>
> 4. floating-point neuromorphic platform
>
> The experimental data presented in the paper is implemented in a GPU-based floating-point computing environment. We have implemented the surrogate gradient using PyTorch's autograd and custom gradient functions.
>
> Additionally, SOLO attempts to maintain the conventional SNN layout, which includes spike generators (or spike routers), weights, and a neuron core. Additional computational components such as pPLIF neurons, pPLI neurons (equivalent to spike counters), and surrogate strategies are designed for ease of hardware implementation. Therefore, if your platform has a conventional SNN layout and you have access to computing components for weight update, SOLO can function well on a floating-point-based neuromorphic platform, both automatically and manually. We are also very interested in digital neuromorphic platforms and are actively working to deploy SOLO on a board system. If you need any further information, we would be happy to assist.

---

### Official Review · Reviewer_TEM3 · 2023-10-26

**Soundness:** 3 good
**Presentation:** 3 good
**Contribution:** 2 fair
**Rating:** 5
**Confidence:** 4

**Summary:**

This paper proposes an online learning method for SNNs. A spiking neuron layer without firing is used to accumulate outputs. Then four surrogate strategies are proposed:
1. Using a boxcar surrogate function with only a 0/1 gradient.
2. Using an always-on gradient in loss.
3. Redefining the gradient of max pooling to propagate gradients to those elements that are not the maximum values in the pooling windows.
4. Using eligible traces to calculate gradient online.

The proposed methods are validated on some popular datasets.

**Strengths:**

As an online learning method, this paper achieves O(1) memory complexity, which is meaningful for the SNN community.

The proposed method is hardware-friendly and has the potential to be applied to neuromorphic chips.

**Weaknesses:**

The accuracy drops sharply in all datasets except for the toy MNIST dataset, which can not show the effectiveness of the proposed methods. I am afraid that the plain SNN with a simple Real Time Recurrent Learning method will get close performance to the proposed methods.

As a comparison, OTTT [1] is also an online training method and achieves much higher accuracy even on the challenging ImageNet dataset.

[1] Xiao, Mingqing, et al. "Online training through time for spiking neural networks." Advances in Neural Information Processing Systems 35 (2022): 20717-20730.

**Questions:**

I do not understand the necessity of "surro2: Always-On beyond Spike Accumulation". The authors claim that they "ensuring error propagation across all classes". But the gradient to each class is not zero in most cases unless the neuron that represents a class outputs 0 at all time-steps when it is not the target class (or outputs 1 at all time-steps when it is the target class).

---

> ### Author Response · Authors · 2023-11-17
> **Initial Response to Reviewer TEM3**
>
> First and foremost, I would like to express my gratitude for your insightful comments and suggestions.
>
> Please check above 'Initial Response to Reviewer Comments'.
>
> 1. Accuracy
>
> We aim to compare the performance of learning algorithms using the same training configuration. Therefore, we use the results shown in AutoSNN as a baseline. Additionally, we compare BPTT, STBP, and SOLO under the same training configuration, which includes the utilization of the pPLIF neuron model, the accumulative neuron model, regularization, etc.
>
> However, unlike the regularization technique used in AutoSNN, we do not employ spike regularization [1]. This approach involves counting all spikes and incorporating them as an additional term in the weight update equation, which requires extra hardware operations and implementation. Instead, SOLO relies on the regularization effect of the accumulative neuron model, which operates similarly to a neuron. This method does not require any additional regularization term in the weight update equation, making it more advantageous for hardware implementation.
>
> Furthermore, AutoSNN is originally composed of 13 convolutional layers and 1 fc layer, and through light-weighting, it is configured with 11 convolutional layers. (See Appendix Figure 4) However, recent state-of-the-art (SOTA) models employ bigger models such as ResNet 18, 34, 50, and others; hence, they are expected to exhibit higher accuracy on large datasets like CIFAR-100 and ImageNet.
>
> [1] Pellegrini, et al. Low-activity supervised convolutional spiking neural networks applied to speech commands recognition. 2021 IEEE Spoken Language Technology Workshop.
>
> 2. surro2: Always-On beyond Spike Accumulation
>
> First, we calculate the loss based on the spike counting method by counting the output spikes of the network at each time step. In conventional SNNs, like STBP, a'spike counter’ performs this role, and in SOLO, a component named 'accumulative neuron' serves this purpose.
>
>  In STBP, where backpropagation is conducted for information on all time steps, every output spike uniformly propagates the error across classes (see Appendix Figure 3). However, in SOLO, backpropagation is only conducted for information about the final time step, meaning that classes without output spikes in the final time step do not receive error propagation. To address this, we employ 'Always-On beyond Spike Accumulation'.
>
> As you mentioned, at the final time step, the gradient for each class is not zero, even if it is not the target class. It is true. However, the values for the non-target class are involved in the computation of the mean squared error (MSE) loss with the target vectors, leading to negative error, which is appropriately integrated into the network. As a result, this process prevents the non-target class from being output. Additionally, constantly sending the potential of all time steps aids in achieving a learning curve for SOLO that is similar in speed to STBP (see Figure 2). Furthermore, this approach is also beneficial for hardware operations and implementation.
> Additionally, the results of the ablation study for surrogate strategies, conducted as described in Appendix B.5, are documented. In the case of AW (Always-On beyond Spike Accumulation), the best accuracy is observed.

---

> > ### Comment · Reviewer_TEM3 · 2023-11-21
> >
> > Hi, the low performance of your SNNs may be caused by the using of a small network structure. Can you provide the performance of using SOLO on the same structure from previous SOTA research?

---

> ### Author Response · Authors · 2023-11-21
> **Official Comment to Reviewer TEM3**
>
> Hi,
>
> The results of SOLO utilizing various architectures are detailed in Appendix C.6. Utilizing the Resnet19-Zheng network [1], SOLO achieved a 65.72% accuracy on the CIFAR-100 dataset, compared to 57.86% accuracy when employing AutoSNN. This demonstrates that as the network grows, the accuracy sufficiently increases. However, our paper primarily aims to compare the performance of learning algorithms under the same training configuration and specifically using the AutoSNN network.
>
> Additionally, as you noted, we are currently experimenting with the Resnet18 architecture, which incorporates pre-activation residual blocks and has already been implemented in the SLTT model. [2] (It is expected that the ResNet18 model will yield higher results by using pre-activation residual blocks.) The coding for this model is complete, and it is now being tested on various datasets without some details (such as regularization, etc.). This process may take some time, but we will share the results as soon as they become available.
>
> [1] Hanle Zheng, et al. Going deeper with directly-trained larger spiking neural networks. Proceedings of the AAAI Conference on Artificial Intelligence, 10, 2021.
>
> [2] Qingyan Meng, et al. Towards memory- and time-efficient backpropagation for training spiking neural networks, 2023a.

---

> ### Author Response · Authors · 2023-11-23
> **Official Comment to Reviewer TEM3**
>
> We conduct experiments with SOLO utilizing ResNet18 without any regularization, such as dropout.  We achieve accuracy
>  of 92.21% on CIFAR-10 (AutoSNN - 91.33% / resnet19 - 91.10%) and 70.25% on CIFAR-100. (AutoSNN - 57.86% / resnet19 - 65.72%).
>
> Although not yet optimized, the training configuration remains the same as the original one. (See details in Appendix C.4)

---

> > ### Comment · Reviewer_TEM3 · 2023-11-23
> >
> > Thanks for the latest result. I am willing to raise my score.

---

### Official Review · Reviewer_gzEk · 2023-10-30

**Soundness:** 1 poor
**Presentation:** 2 fair
**Contribution:** 2 fair
**Rating:** 3
**Confidence:** 4

**Summary:**

This paper proposes Surrogate Online Learning at Once (SOLO) for training SNNs in a hardware-friendly manner. It only leverages spatial gradient at the final time step for low computational complexity. Experiments are conducted on static and neuromorphic datasets to verify the effectiveness.

**Strengths:**

This paper considers online SNN training methods to promote online on-chip learning, which is an important topic.

**Weaknesses:**

1. The motivation to only consider the gradient at the last time step is not convincing enough, and the experimental results are quite poor. There is no formal/theoretical justification for the claim “we believe that the information of the accumulative neurons in the final time step could yield the most distinct and clear error value among all given time steps”. It is obvious that only considering spatial gradient for the last time step will lose a lot of information on previous time steps, and the experimental results indeed show a significant drop in accuracy, especially for neuromorphic datasets with temporal sequences. For static datasets, there is no temporal information and binary neural networks (or taking T=1 for SNNs) can easily work well, so experiments are not surprising or appealing. It is unclear what’s the advantage of the proposed method over existing online training methods [1,2].

2. The idea of the proposed method may, to some extent, be viewed as a special case of a recent work [3]. It proposes a method SLTT, which drops the temporal dependency of BPTT and only uses spatial gradients at each time step, and it further proposes a variant SLTT-k, which randomly samples k time steps for the spatial gradient. The method in this paper may be viewed as taking k=1 and fixing the considered time step as the last time step. However, this paper ignores gradients for previous time steps, leading to much poorer performance.

[1] A solution to the learning dilemma for recurrent networks of spiking neurons. Nature Communications, 2020.

[2] Online Training Through Time for Spiking Neural Networks. NeurIPS, 2022.

[3] Towards Memory- and Time-Efficient Backpropagation for Training Spiking Neural Networks. ICCV, 2023.

**Questions:**

1. It is not clear enough why pPLIF is more straightforward for hardware implementation than PLIF. If consider deploying trained models, $\beta$ for the current in PLIF can be absorbed into the weight. If consider training models, it is also unclear for pPLIF how the gradient for the learnable membrane time constant can be calculated on hardware.

---

> ### Author Response · Authors · 2023-11-17
> **Initial Response to Reviewer gzEk**
>
> First and foremost, I would like to express my gratitude for your insightful comments and suggestions.
>
> Please check above  'Initial Response to Reviewer Comments'.
>
> 1. Similar to SLTT
>
> We have reviewed the SLTT paper, and as you mentioned, its intent is somewhat similar to SOLO, and SOLO can be considered a special case of SLTT-K. However, SLTT-K performs backpropagation for randomly sampled K time steps at each training point, and there's no results of the case where K = 1 (minimum K = 2). Furthermore, SLTT-K ultimately requires memory to hold spatial states for specific K-time steps, making O(1) memory complexity impossible on neuromorphic platforms. In contrast, SOLO has the advantage of being able to directly access all components for weight updates at the final time step, transitioning to the weight update phase without the need for memory on neuromorphic platforms. In summary, in our work, SOLO demonstrates performance in harsher training configurations and hardware environments compared to SLTT.
>
> 2. pPLIF and gradient toward the time constant
>
> You can find information on the PLIF neuron model in Appendix A.1, 'Neuron Model.' In the PLIF neuron model, the parameter \beta is shared for use both in the computation of membrane potential and in the modulation of current. \beta for the current couldn’t be ignored (or absorbed to current) when calculating the gradient with respect to \beta. In Appendix A.5, 'Related Gradient Chain,' you can find the process used to calculate the gradient with respect to the membrane constant in PLIF neurons, comparing it with pPLIF neurons and SOLO. On pPLIF neurons and SOLO. In the case of pPLIF neurons and SOLO, the gradient flowing through the time constant is simplified, which is expected to facilitate easier hardware implementation.

---

### Official Review · Reviewer_wijC · 2023-11-06

**Soundness:** 2 fair
**Presentation:** 3 good
**Contribution:** 2 fair
**Rating:** 5
**Confidence:** 5

**Summary:**

This paper proposes a surrogate online learning method-SOLO, to efficiently train spiking neural networks. The main idea is to consider a backward path only at the final step, which disentangles the temporal dependencies of the conventional BPTT-type training method. The authors show that by doing so, the performance on several benchmark tasks does not decrease significantly, while largely reducing the required memory and training time. This shows the potential to be implemented in the neuromorphic hardware in the future.

**Strengths:**

- The paper is well-written, with very clear illustration on the motivations, methods and implementations.
- The paper proposes a new way to efficiently train spiking neural networks, and this method shows the potential to solve the on-chip learning challenge of neuromorphic chips.

**Weaknesses:**

- The paper, however, lacks of a enough investigation and comparison with the existing methods. Aiming to cut the temporal dependencies to optimize the SNN training is not a new idea [ref. 1-3], what are the main differences (except for the “last time step” part) compared with them? For instance, the intrinsic idea, the approximation way, even the three-factor-rules part are quite similar as in [1].
- The resulting performance decreases, if not significantly, still quite a lot, on many datasets. One might suspect the availability of this method in real use cases.

Ref:
1. Online Training Through Time for Spiking Neural Networks, NeurIPS22
2. Towards Memory- and Time-Efficient Backpropagation for Training Spiking Neural Networks, ICCV23
3. Accurate online training of dynamical spiking neural networks through Forward Propagation Through Time, Nature MI

**Questions:**

See the above weakness part.
In addition, in table 1 and table 2, the baseline performance looks not very high, e.g., for CIFAR 10, SNN SOTA is already close to 95% with 4-6 time steps, and CIFAR100 around 73%, but in these tables, these number are relatively low, could you explain the reasons?

---

> ### Author Response · Authors · 2023-11-17
> **Initial Response to Reviewer wijC**
>
> First and foremost, I would like to express my gratitude for your insightful comments and suggestions.
>
> Please check above 'Initial Response to Reviewer Comments'.
>
> 1. Three-factor-rule and OTTT
>
> Since OTTT is also based on backpropagation, its three-factor rules may be somewhat similar to those of SOLO.
>
> 2. accuracy
>
>  We aim to compare the performance of learning algorithms using the same training configuration. Therefore, we use the results shown in AutoSNN as a baseline. Additionally, we compare BPTT, STBP, and SOLO under the same training configuration, which includes the utilization of the pPLIF neuron model, the accumulative neuron model, regularization, and etc.
>
>  However, unlike the regularization technique used in AutoSNN, we do not employ spike regularization [1]. This approach involves counting all spikes and incorporating them as an additional term in the weight update equation, which requires extra hardware operations and implementation. Instead, SOLO relies on the regularization effect of the accumulative neuron model, which operates similarly to a neuron. This method does not require any additional regularization term in the weight update equation, making it more advantageous for hardware implementation.
>
> Furthermore, AutoSNN is originally composed of 13 convolutional layers and 1 fc layer, and through light-weighting, it is configured with 11 convolutional layers. (See Appendix Figure 4) However, recent state-of-the-art (SOTA) models employ bigger models such as ResNet 18, 34, 50, and others; hence, they are expected to exhibit higher accuracy on large datasets like CIFAR-100 and ImageNet.
>
>
> [1] Pellegrini, et al. Low-activity supervised convolutional spiking neural networks applied to speech commands recognition. 2021 IEEE Spoken Language Technology Workshop.

---

### Author Response · Authors · 2023-11-17
**Initial Response to Reviewer Comments.**

We notice that there are several additional explanations in the appendix for the reviewer.

C.6. Results of SOLO utilizing various architectures

C.11. Cross Bar Array Application of SOLO for Analog Synapses

C.12. Comparison of Recent Efficient Training Methods for SNNs

1.

As mentioned in the introduction, the concept of spiking neural networks (SNNs) is suitable for low-power and hardware implementations, mimicking the biological brain. However, recent works on SNNs have focused more on performance than on hardware-friendly learning algorithms and computational components. SOLO is designed with a focus on hardware implementation and could demonstrate suitable performance in terms of accuracy, time/memory complexity, and hardware scenarios. We utilize the network of AutoSNN, which is the NAS of SNNs. [1] AutoSNN is originally composed of 13 convolutional layers and 1 fc layer, and through light-weighting, it is configured with 11 convolutional layers. (See Appendix Figure 4) When compared to other learning algorithms, the baseline and SNNs implemented with SOLO may not represent the highest performance on some large datasets because of differences in networks and additional computing components.

2.

In the field of neuromorphic algorithms, interpreting biological evidence and co-designing it with existing spiking neural network (SNN) systems are crucial. An example of this is feedback alignment [2], which was proposed focusing on the biological and hardware-friendly idea, and its mathematical background was researched after the proposal. [3] And the idea of removing the temporal gradient is referenced in SOLO [4], and it aligns with the approaches of other algorithms like OTTT and SLTT. In our work, however, we've reflected on an idea based on biological evidence [5, 6] (refered in SOLO), expecting that it would be suitable for hardware operations, emphasizing minimal computational and memory demands, as well as hardware-specific noise and reliability concerns, much like an actual brain. Surprisingly, it showed little difference from conventional training algorithms under the same training configuration. For more on the mathematical background, it would be appreciated if you could refer to the appendix material A.1.~B.2..

3.

SOLO, conducting backpropagation only at the final time step, indeed does not directly access the temporal information of previous time steps. However, due to the recurrent nature of SNNs, the network of SOLO still indirectly reflects temporal information from previous steps through the potential of the accumulative neuron, the potential of membrane potential, and the occurrence of spikes at the final time step. [7] Consequently, the information of the final time step determines the candidates for weight updates by reflecting the previous temporal data with pPLIF neurons, accumulative neurons, and surrogate strategies. Therefore, this is partly why satisfactory results from SOLO are obtained in classification tasks involving neuromorphic datasets.

(When using T = 1 for the CIFAR-10 dataset, the results are as follows: For SOLO, the training accuracy is about 90% (90.35%), but the test accuracy is around 50% (52.21%). For SOLOet, the training accuracy is about 90% (88.67%), while the test accuracy is around 50% (48.84%))

4.

Additionally, we include results utilizing various architectures (the result of SOLO utilizing the Resnet19-Zheng network is 65.72% on CIFAR-100), a real application scenario of SOLO, such as the cross bar array application, and a table comparing various recent works. Therefore, it would be helpful to check for the above additional Appendix materials.

[1] Na, et al. Autosnn: Towards energy-efficient spiking neural networks, 2022 ICLR.

[2] Lillicrap, T., et al. Random synaptic feedback weights support error backpropagation for deep learning. 2016, Nat Commun.

[3] Maria Refinetti, et al. Align, then memorise: the dynamics of learning with feedback alignment. 2022, J. Stat. Mech.

[4] Wachirawit Ponghiran and Kaushik Roy. Spiking neural networks with improved inherent recurrence dynamics for sequential learning. 2022 AAAI.

[5] Birte U. Forstmann, et al. Cortico-striatal connections predict control over speed and accuracy in perceptual decision making. 2010, Proceedings of the National Academy of Sciences

[6] Jie Mei, Eilif Muller, and Srikanth Ramaswamy. Informing deep neural networks by multiscale principles of neuromodulatory systems. 2022, Trends in Neurosciences.

[7] Kim, et al. Exploring  Temporal Information  Dynamics in Spiking Neural Networks. 2023, AAAI

---

### Meta-Review · Area_Chair_mueR · 2023-12-11

**Metareview:**

The paper introduces SOLO, a training algorithm for Spiking Neural Networks (SNNs) designed for hardware efficiency. SOLO conducts backpropagation only at the final time step to reduce memory and computational requirements. The authors argue that this approach is suitable for low-power, hardware-based implementations of SNNs. The paper presents experimental results on various datasets to support these claims.

The reviewers noted the following weaknesses, which were not fully addressed in the rebuttal stage
1. The paper lacks a strong theoretical foundation for the proposed method, relying primarily on empirical results.
2. Lack of comprehensive comparison with existing methods, especially with other online training methods for SNNs.
3. The reported performance, particularly on complex datasets, does not convincingly outperform existing methods.
4. Some reviewers noted that the paper lacks clarity in its mathematical expressions and justifications.
5. While the paper emphasizes hardware implementation, it does not sufficiently address how the algorithm would be implemented on actual neuromorphic hardware.

**Justification For Why Not Higher Score:**

While the paper introduces a potentially impactful concept in the field of SNNs, it falls short in providing a convincing theoretical framework, comprehensive comparative analysis, and clear evidence of practical superiority over existing methods. The concerns regarding mathematical clarity and details of hardware implementation further weaken the paper's standing.

**Justification For Why Not Lower Score:**

N/A

---

### Decision · Program_Chairs · 2024-01-16

Reject